

# Late Cretaceous sauropod tooth morphotypes may provide supporting evidence for faunal connections between North Africa and Southern Europe

Femke M. Holwerda[1,2,3], Verónica Díez Díaz[4,5], Alejandro Blanco[2,6], Roel Montie[1,2,3] and Jelle W.F. Reumer[1]

[1] Faculty of Geosciences, Utrecht University, Utrecht, Netherlands
[2] SNSB-Bayerische Staatssammlung für Paläontologie und Geologie, Munich, Bavaria, Germany
[3] Department of Earth and Environmental Sciences and GeoBioCenter, Ludwig-Maximilians-Universität München, Munich, Bavaria, Germany
[4] Museum für Naturkunde, Leibniz-Institut für Evolutions-und Biodiversitätsforschung, Berlin, Germany
[5] Humboldt Universität Berlin, Berlin, Germany
[6] Centro de Investigacións Científicas Avanzadas (CICA), Facultade de Ciencias, Universidade da Coruña, A Coruña, Spain

Corresponding author
Femke M. Holwerda,
f.holwerda@lrz.uni-muenchen.de,
f.m.holwerda@gmail.com

## ABSTRACT

The Cretaceous Kem Kem beds of Morocco and equivalent beds in Algeria have produced a rich fossil assemblage, yielding, amongst others, isolated sauropod teeth, which can be used in species diversity studies. These Albian-Cenomanian (∼113–93.9 Ma) strata rarely yield sauropod body fossils, therefore, isolated teeth can help to elucidate the faunal assemblages from North Africa, and their relations with those of contemporaneous beds and geographically close assemblages. Eighteen isolated sauropod teeth from three localities (Erfoud and Taouz, Morocco, and Algeria) are studied here, to assess whether the teeth can be ascribed to a specific clade, and whether different tooth morphotypes can be found in the samples. Two general morphotypes are found, based on enamel wrinkling and general tooth morphology. Morphotype I, with mainly rugose enamel wrinkling, pronounced carinae, lemon-shaped to (sub)cylindrical cross-section and mesiodistal tapering towards an apical tip, shows affinities to titanosauriforms and titanosaurs. Morphotype II, characterized by more smooth enamel, cylindrical cross-section, rectangular teeth with no apical tapering and both labial and lingual wear facets, shows similarities to rebbachisaurids. Moreover, similarities are found between these northwest African tooth morphotypes, and tooth morphotypes from titanosaurs and rebbachisaurids from both contemporaneous finds from north and central Africa, as well as from the latest Cretaceous (Campanian–Maastrichtian, 83.6 Ma–66.0 Ma) of the Ibero-Armorican Island. These results support previous hypotheses from earlier studies on faunal exchange and continental connections between North Africa and Southern Europe in the Cretaceous.

## INTRODUCTION

The early Late Cretaceous of northwestern Africa is well-known for its rich vertebrate fauna, many taxa having been described in particular from the Albian–Cenomanian (~113–93.9 Ma) Kem Kem beds of Morocco, and the Albian–Cenomanian equivalent continental intercalaire of Algeria. The Moroccan Kem Kem beds include aquatic fauna such as sharks, lungfish, coelacanths, bony fish, amphibians, turtles, crocodylomorphs, as well as terrestrial vertebrates such as squamates, pterosaurs, sauropods, and an abundance of theropods (*Lavocat, 1954*; *Russell, 1996*; *Sereno et al., 1996*; *Wellnhofer & Buffetaut, 1999*; *Cavin et al., 2010*; *Richter, Mudroch & Buckley, 2013*; *Läng et al., 2013*; *Mannion & Barrett, 2013*). Despite this large diversity, most fossil material consists of isolated elements from theropods and chondrichthyans (e.g., *Spinosaurus, Carcharodontosaurus, Onchopristis*, *Läng et al., 2013*; C Underwood, pers. comm., 2018). *Läng et al. (2013)* attributed this to the deltaic palaeoenvironment being unsuitable for the setting of stable terrestrial vegetation. Because of this, the herbivorous fauna has not received much attention thus far, and sauropod material is rare (C Underwood, pers. comm., 2018, but see *McGowan & Dyke, 2009*). Studies of sauropod material from this region thus far found *Rebbachisaurus garasbae*, and other rebbachisaurids (*Lavocat, 1954*; *De Lapparent & Gorce, 1960*; *Russell, 1996*; *Mannion & Barrett, 2013*; *Wilson & Allain, 2015*) as well as several titanosauriform remains, and also a possible titanosaurian (*De Broin, Grenot & Vernet, 1971*; *Kellner & Mader, 1997*; *Mannion & Barrett, 2013*; *Lamanna & Hasegawa, 2014*; *Ibrahim et al., 2016*). *De Lapparent & Gorce (1960)* also mentioned brachiosaurid finds, however, these remains are now considered to be rebacchisaurid or titanosauriform (*Mannion, 2009*; *Mannion & Barrett, 2013*).

Sauropod body fossils are restricted to mostly isolated elements, or, if associated, material is not as numerous as with theropod material (see e.g., *Mahler, 2005*; *Novas, Dalla Vecchia & Pais, 2005*; *Cau & Maganuco, 2009*). Sauropod teeth, however, are preserved in relative abundance. One isolated sauropod tooth had already been reported on by *Kellner & Mader (1997)*. Sauropod teeth are commonly preserved in the fossil record due to their hardness, resilience against weathering, and due to their high tooth replacement rates (see e.g., *Calvo, 1994*; *Erickson, 1996*; *García & Cerda, 2010*; *García & Cerda, 2010b*). Studying isolated teeth has previously been applied to assessing theropod species diversity in North Africa (*Richter, Mudroch & Buckley, 2013*). Sauropod teeth can be used for a similar purpose as well, as morphological classifications based on shape, size and position of wear facets (*Calvo, 1994*; *Salgado & Calvo, 1997*; *Chure et al., 2010*; *Mocho et al., 2016*; *Carballido et al., 2017*), and enamel wrinkling patterns (*Carballido & Pol, 2010*; *Díez Díaz, Pereda Suberbiola & Sanz, 2012*; *Díez Díaz, Tortosa & Loeuff, 2013*; *Holwerda, Pol & Rauhut, 2015*) have classified tooth assemblages into morphotypes or even down to family or genus level (e.g., *Amygdalodon, Patagosaurus*).

*Mannion & Barrett (2013)* suggested that the Cretaceous North African titanosauriforms may not be closely related to southern African forms, as the lineages were cut off from each other by the trans-Saharan seaway. Moreover, close relations are suggested between Cretaceous North African sauropods and Italian sauropods (*Zarcone et al., 2010*; *Dal*

*Sasso et al., 2016*), and Iberian sauropods (*Sallam et al., 2018*; *Díez Díaz et al., 2018*). More specifically, close relations between Egyptian and European sauropods (*Sallam et al., 2018*) and between Tunisian and European sauropods (*Fanti et al., 2015*) have been found. These studies proposed faunal exchanges during the Late Cretaceous between northern Africa and southern Europe. Several migratory routes have been suggested, such as the 'Apulian route' during the Early Cretaceous (*Dalla Vecchia, 2002*; *Canudo et al., 2009*). Continental connections would have been made possible by peri-Adriatic carbonate platforms in the Mediterranean, connecting North Africa with Adria, throughout the Cretaceous, making migration possible between the northern African and southern European islands and peninsulas (*Zarcone et al., 2010*). Indeed, these carbonate platforms contain numerous tetrapod footprints, including those of sauropods (*Zarcone et al., 2010*). The hypothesis of a faunal exchange during the Cretaceous is not new; Late Cretaceous abelisaurid theropods and titanosaurian sauropods from France are found to have Gondwanan affinities, indicating migration from Gondwana to Europe, an event which could already have taken place in the Early Cretaceous (*Buffetaut, Mechin & Mechin-Salessy, 1988*; *Buffetaut, 1989*). Next to sauropods, Early Cretaceous abelisaurid and carcharodontosaurid theropods were found with Gondwanan affinities, as well as other terrestrial fauna, such as amphibians, snakes, and ziphodont crocodyliforms (*Le Loeuff, 1991*; *Vullo et al., 2005*; *Vullo, Neraudeau & Lenglet, 2007*; *Pereda-Suberbiola, 2009*). *Ösi, Apesteguía & Kowalewski (2010)* found Santonian theropods from the Mediterranean region to have both Gondwanan and North American affinities. *Dalla Vecchia & Cau (2011)* added a notosuchian from the Late Cretaceous of Italy, and *Rabi & Sebök (2015)* a sebecosuchian to this faunal assemblage with Gondwanan affinities. Reviewing undescribed North African Cretaceous sauropod material could add information on both the biogeographical patterns of the Euro-Gondwanan area, as well as on sauropod species diversity in northwestern Africa.

Here, we present a morphological and quantitative analysis of a sauropod tooth assemblage from the Cenomanian of Morocco and Algeria. Teeth are categorized into two morphotypes, which are then compared to contemporaneous Cretaceous sauropod tooth morphotypes, including sauropod teeth from Africa and southern Europe.

## GEOLOGICAL SETTING

Fourteen of the teeth studied here are from the Kem Kem beds of Morocco, and four were supposedly found in the Late Cretaceous Continental Intercalaire of Algeria. Four of the Moroccan sample are labeled as originating from 'Taouz, Algeria,' and four are labeled 'Kem Kem Morocco'. Taouz, Algeria, might actually mean Taouz, Morocco, which is the southern part of the Kem Kem beds, and ten samples are labeled as originating from Erfoud, Morocco, which is the more northern part of the Kem Kem beds (Fig. 1). The Kem Kem area is located in the south-east of Morocco (Fig. 1). Here, the Kem Kem beds form an escarpment around the eastern end of the Anti-Atlas, from near Goulmima in the northwest (C Underwood, pers. comm., 2018; see Fig. 1) past Erfoud in the north (*Wellnhofer & Buffetaut, 1999*; *Cavin & Forey, 2004*) and along the east, parallelling the Algerian border (C Underwood, pers. comm., 2018). The Kem Kem ends to the west, south

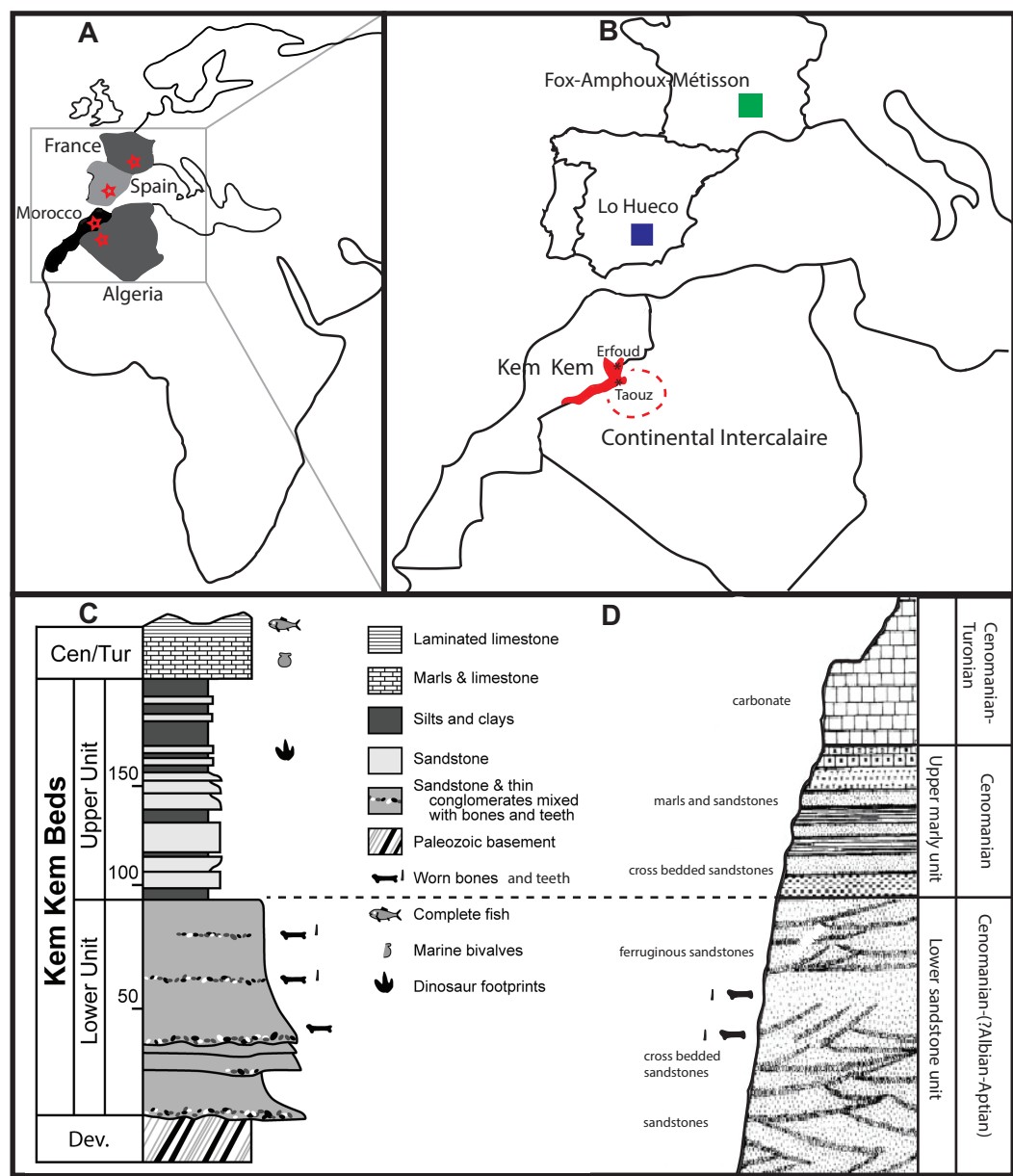

**Figure 1** **Geological setting of Kem Kem beds, Morocco, and Continental Intercalaire, Algeria.** Taouz and Erfoud are indicated. localities of Lo Hueco, Spain, and France (FAM, AIX, Massecaps) are also portrayed. (A) Map of Europe and Africa. (B) Close-up of North Africa and Southern Europe. (C) stratigraphical column of Kem Kem beds (after *Belvedere et al., 2013*; *Ibrahim et al., 2014*). (D) stratigraphical column of Continental Intercalaire, Algeria, (after *Forey, López-Arbarello & MacLeod, 2011*).

of Taouz, thereby in total stretching to about 300 km of outcrop (C Underwood, pers. comm., 2018; see Fig. 1). The Kem Kem is usually mentioned to be Cenomanian in age, as it has been found to match ammonites from the Lower Cenomanian of Bahariya, Egypt (*Le Loeuff et al., 2012*). However, the age range could span over the Albian–Cenomanian (C Underwood, pers. comm., 2018), which is closer to the age given to the Algerian

Cretaceous Continental Intercalaire (*Lefranc & Guiraud, 1990*; *Le Loeuff et al., 2012*) and to the fossil-rich Cretaceous 'Continental Intercalaire' beds of Tunisia (*Fanti et al., 2016*) as well as sauropod bonebeds from Niger (*Sereno & Wilson, 2005*). The Kem Kem beds are considered to be made up of two formations (see Fig. 1): the fossil-rich lower Ifezouane Formation and the upper Aoufous Formation, rich in ichnofossils (*Cavin et al., 2010*; *Belvedere et al., 2013*); also named the lower sandy unit (a braided fluvial system) and the upper marly unit (a coastal lagoon), respectively (*Cavin et al., 2010*; *Belvedere et al., 2013*; *Mannion & Barrett, 2013*; *Ibrahim et al., 2014*). Practically all fossil vertebrates originate from the lower Ifezouane Formation (*Cavin et al., 2010*, C Underwood, pers. comm., 2018). The Continental Intercalaire of Algeria is less studied than the Kem Kem, and the age ranges from Barremian to Turonian. However, most authors set the age of the beds close to the Moroccan border (where our Algerian specimens are supposedly from), to Albian-Cenomanian-Turonian, with the Cenomanian layers being the most fossil-rich (*Läng et al., 2013*; *Benyoucef et al., 2015*; *Meister et al., 2017*). As said before, the labeling, however, of 'Taouz, Algeria' is most likely not correct, as Taouz is situated in Morocco, south of Erfoud, close to the border with Algeria, where many fossils from the southern Kem Kem exposures, towards Ouzina, are collected over a broad expanse, usually in mines around Bagaa (M Dale, C Underwood, pers. comm., 2018; Fig. 1). Indeed, Taouz, Morocco, is indicated as fossil locality in other Kem Kem fossil vertebrate studies (e.g., *Wellnhofer & Buffetaut, 1999*; *Cavin et al., 2010*; *Forey, López-Arbarello & MacLeod, 2011*; *Richter, Mudroch & Buckley, 2013*). As the Kem Kem outcrops at present run parallel to the border, the labelling of 'Algeria' is probably still be correct, as the 'Kem Kem' beds extend out across the border (*Alloul et al., 2018*), however, the specific provenance of these teeth is unclear.

Next to the labelling, the colour of the fossils likely confirms the provenance on the labels, as fossils from Bagaa (Taouz) are chocolate brown in colour, and fossils from north of Erfoud show a range of colours, usually shades of beige and black (C Underwood, pers. comm., 2018), matching the provenance on the collection reference (see 'Description').

As most of the fossils retrieved from the Kem Kem and equivalent beds from Algeria are found via mining, the provenance is unfortunately usually unclear and only traceable to a regional provenance (*Forey & Cavin, 2007*; *Rodrigues et al., 2011*; C Underwood, pers. comm., 2018).

## MATERIALS AND METHODS

In this study, eighteen isolated sauropod teeth from the Kem Kem beds and Continental Intercalaire are studied for shape, size, position of wear facets (where applicable) and enamel wrinkling. The four teeth from around Taouz (Morocco) are BSPG 1993 IX 331A, BSPG 1993 IX 331B, BSPG 1993 IX 331C, and BSPG 1993 IX 313A, see Figs. 2A–2D. The ten teeth from Erfoud (Morocco) are labeled PIMUZ A/III 0823, and are given the additional labeling of a, b, c, etc., for convenience, see Figs. 3A–3J. The four Algerian specimens are BSPG 1993 IX 2A, BSPG 1993 IX 2B, BSPG 1993 IX 2C, and BSPG 1993 IX 2D, see Figs. 2E–2H. Measurements were taken with a caliper to mm scale. For imaging, the

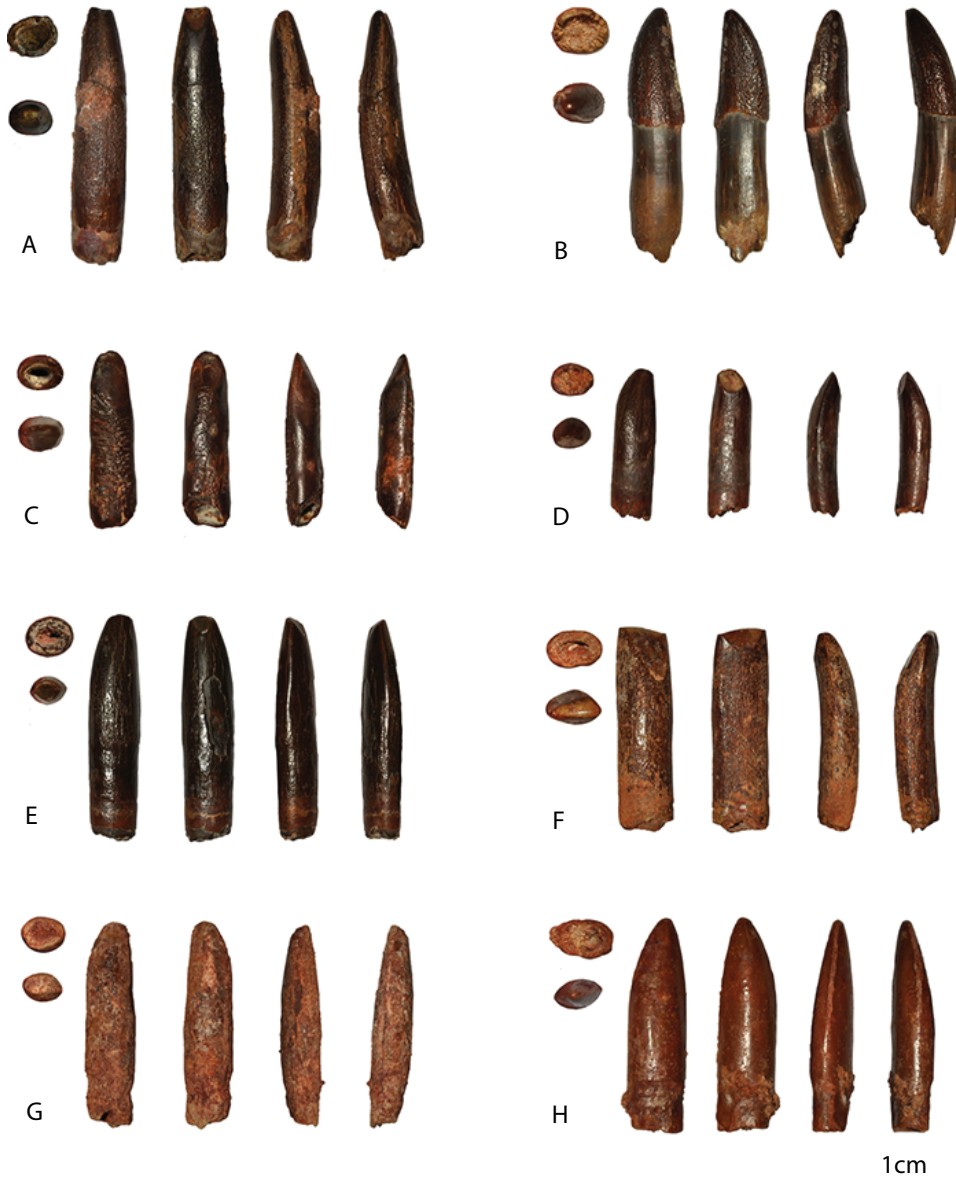

**Figure 2** **Images of BSPG 1993 IX 331A (A), BSPG 1993 IX 331B (B), BSPG 1993 IX 331C (C), BSPG 1993 IX 313A (D), BSPG 1993 IX 2A (E), BSPG 1993 IX 2B (F), BSPG 1993 IX 2C (G), and BSPG 1993 IX 2D (H).** In basal view, apical view, labial view, lingual view, distal view, and mesial view. The scale bar equals 1 cm. Images taken by RM.

teeth were photographed using normal and macro settings. Scanning Electron Miscroscopy (SEM) pictures were taken of the Munich sample at the Zoologisches Institut in Munich to obtain a detailed view of the enamel wrinkling patterns. SEM images were not possible to obtain for the Zürich sample. The specimens were examined using a LEO 1430VP SEM. In this study, the proposed dental orientations of *Smith & Dodson (2003)* are followed. The Slenderness Index (SI, sensu, *Upchurch, 1998*) was measured for each crown tooth

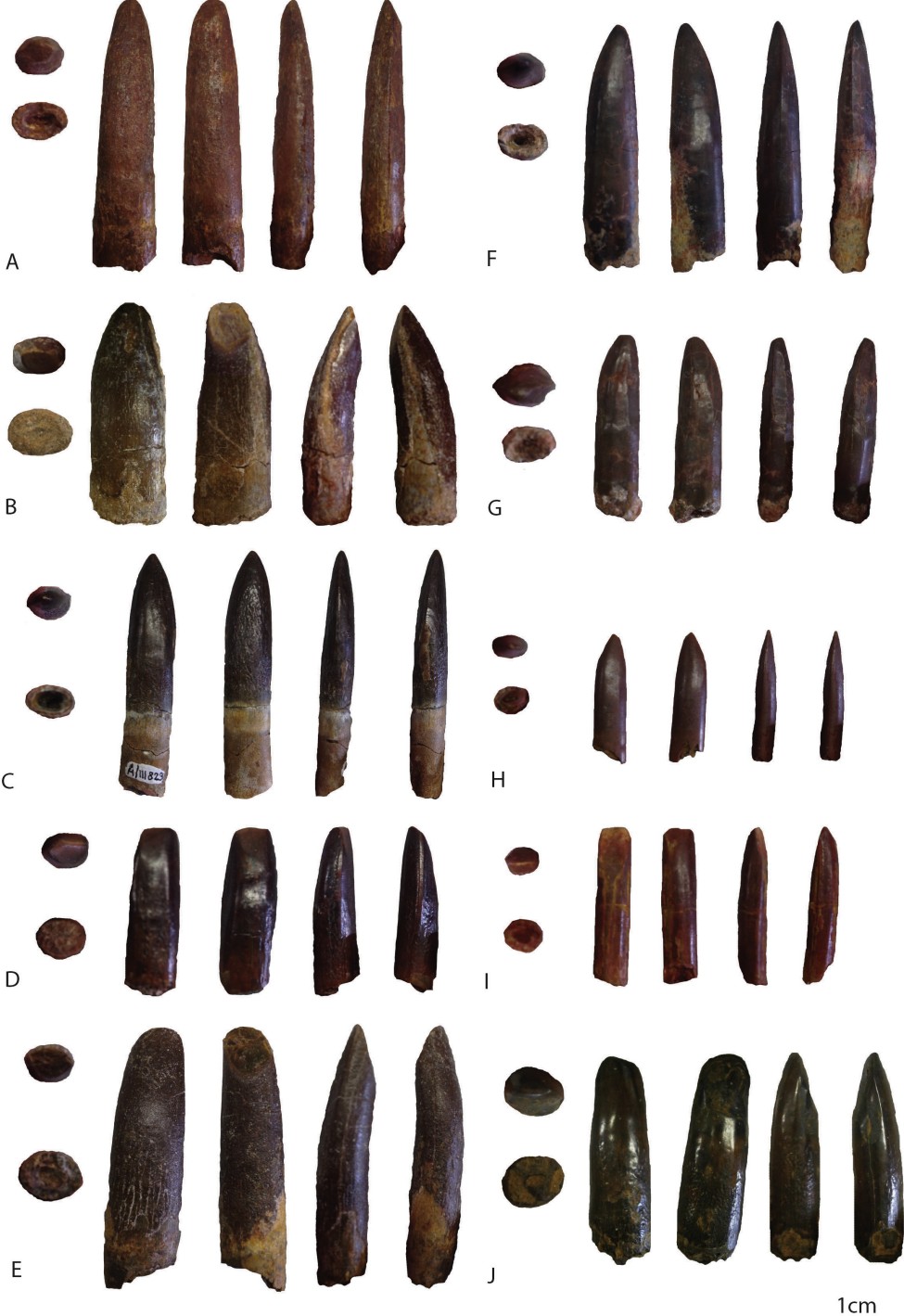

**Figure 3  Images of PIMUZ A/III 0823a (A), PIMUZ A/III 0823b (B) PIMUZ A/III 0823c (C), PIMUZ A/III 0823d (D), PIMUZ A/III 0823e (E), PIMUZ A/III 0823f (F), PIMUZ A/III 0823g(G), PIMUZ A/III 0823 h (H), PIMUZ A/III 0823i (I), PIMUZ A/III 0823j (J).** In apical view, basal view, labial view, lingual view, distal view, and mesial view. The scale bar equals 1 cm. Images taken by FH.

by dividing the apicobasal length by the mesiodistal width in the middle of the crown. The Compression Index (CI, sensu *Díez Díaz, Tortosa & Loeuff, 2013*) was measured for each tooth by dividing the labiolingual width by the mesiodistal width in the middle of the crown. The abrasion stages of the functional dentition proposed by *Saegusa & Tomida (2011)* and *Wiersma & Sander (2016)* will be assessed for each tooth in order to assess possible tooth position in the toothrow. The angles of the wear facets were measured with respect to the labiolingual axes of the teeth. See Table 1 for all tooth measurements.

The studied sample was compared with teeth from other sauropod taxa in a quantitative approach by multivariate analysis. The comparative tooth sample is measured by first hand observations (i. e., the Lo Hueco, Massecaps, Fox-Amphoux-Métisson samples, *Ampelosaurus, Atsinganosaurus, Lirainosaurus, Patagosaurus, Lapparentosaurus*) as well as from literature, where SI and CI were reported or could be confidently measured or estimated, and a minimum sample size of three tooth specimens was reached (*Upchurch & Barrett, 2000*; *Barrett et al., 2002*; *Carpenter & Tidwell, 2005*; *Apesteguía, 2007*; *Freire, Medeiros & Lindoso, 2007*; *Díez Díaz et al., 2012a*; *Díez Díaz, Pereda Suberbiola & Sanz, 2012*; *Díez Díaz, Tortosa & Le Loeuff, 2013*; *Díez Díaz, Ortega & Sanz, 2014*; *Holwerda, Pol & Rauhut, 2015*; *França et al., 2016*; *Averianov & Sues, 2017*). Taxa from different periods and palaeobiogeographic origins were included where possible. A total of 102 teeth were grouped by taxon, if possible, or sorted by morphotype, finally creating 17 different groups. Both damaged/worn as well as unworn teeth were used, due to paucity of sample size. See Table S1 for this data. Differences in SI and CI ratios were tested amongst groups through statistical analyses. The shape of the cross section and the number, angle and size of the wear facets were not considered for this purpose because these features may be more related to other functional factors (i.e., tooth position, stage of tooth wear) rather than a taxonomic factor. Due to the small sample size, the non-normal distribution and the non-homoscedastic variances amongst sample groups, the non-parametric multivariate one-way PERMANOVA test was performed, followed by post-hoc tests assessing differences for each pair of groups (*Hammer & Harper, 2006*). The analysis was implemented using PAST v3.20 (*Hammer, Harper & Ryan, 2001*). Finally, the tooth groups were depicted in a dispersion plot, together with additional taxa for which sample size was too small to be included in the statistical analyses (see 'Discussion').

# RESULTS

## Morphological description
### Moroccan sample: Taouz
*BSPG 1993 IX 331A (Fig. 2A).* The crown of this chocolate-to reddish-brown coloured, apicobasally elongated tooth is more or less cylindrical. It tapers towards the apex, both mesiodistally and labiolingually. The labial side is strongly convex, the lingual side is straight to concave. The convexity increases towards the apex as the distal 1/3rd bends more strongly towards the lingual side. The tooth has an almost circular cross section at the base of the crown, becoming slightly more flattened in the labiolingual direction apically. It has a SI of 4.31 and a CI of 0.85 (see Table 1). Two distinct wear facets are present on

Holwerda et al. (2018), PeerJ, DOI 10.7717/peerj.5925

**Table 1  Measurements, wear stage (after *Saegusa & Tomida, 2011*) and enamel morphology of each tooth.**  Measurements in mm.

| Specimen nr | Apico-basal length | Abrasion stage | Mesio-distal base | Mesio-distal middle | Mesio-distal apex | Labio-lingual base | Labio-lingual middle | Labio-lingual apex | SI | CI | Abrasion stage | Number and location of wear facets | Hypothetical placement in the snout | Apical third cross-section | Enamel ornamentation |
|---|---|---|---|---|---|---|---|---|---|---|---|---|---|---|---|
| BSPG 1993 IX 331A | 60 | 53 | 12 | 12,3 | 6,2 | 10 | 10,5 | 4 | 4,3089 | 0,8536 | F2 | 2 (ap, ln) | Premaxilla | Elliptical | Anastomosed |
| BSPG 1993 IX 331B | 48 | 24 | 10,8 | 11,1 | 4,5 | 8,5 | 8,5 | 3 | 2,1621 | 0,7657 | F1 | 0 | – | Elliptical | Anastomosed |
| BSPG 1993 IX 331C | 37 | 37 | 10 | 9 | 5,5 | 7,5 | 7,5 | 2 | 4,1111 | 0,8333 | F4-F5 | 4 (lb, ln, m, d) | Maxilla | Elliptical | (worn) scratches and apical pits |
| BSPG 1993 IX 313A | 27,5 | 27,5 | 8,2 | 8,2 | 4,5 | 6 | 6,2 | 1 | 3,3536 | 0,7560 | F2 | 1 (ln) | Premaxilla | Elliptical | Longitudinal ridges (but worn) |
| BSPG 1993 IX 2A | 51 | 50 | 10,5 | 11,5 | 5,5 | 8,5 | 9 | 2 | 4,3478 | 0,7826 | F2 | 1 (ln) | Maxilla | Lemon shaped | Longitudinal ridges (but worn) |
| BSPG 1993 IX 2B | 47,5 | 47,5 | 13 | 13 | 11,5 | 9 | 9,2 | 3 | 3,6538 | 0,7076 | F2–F3 | 2 (lb, ln) | Anterior dentary | Lemon shaped | Longitudinal ridges |
| BSPG 1993 IX 2C | 47 | 47 | 10 | 11,5 | 5 | 8 | 10 | 2 | 4,0869 | 0,8695 | F2 | 1 (lb) | Anterior dentary | Lemon shaped | No (because of preservation) |
| BSPG 1993 IX 2D | 38 | 38 | 9,5 | 10,5 | 1,5 | 6,5 | 7,5 | 1,5 | 3,6190 | 0,7142 | F1 | 0 | Anterior (?) | Lemon shaped | (worn) w/pits and scratches |
| PIMUZ A/III 0823 a | 66 | 59 | 15 | 12 | 7 | 8 | 7 | 4 | 4,9166 | 0,5833 | F2 | 1 (ln) | Premaxilla | Elliptical | Anastomosed (but worn) |
| PIMUZ A/III 0823 b | 39 | 31 | 14,5 | 13 | 7 | 8 | 9 | 7 | 2,3846 | 0,6923 | F2 | 1 (ln) | Premaxilla | D-shaped | Anastomosed |
| PIMUZ A/III 0823 c | 57 | 34 | 11 | 9 | 5 | 6 | 6 | 3 | 3,7777 | 0,6666 | F1 | 0 | – | Lemon shaped | Anastomosed w/protrusions |
| PIMUZ A/III 0823 d | 24 | 21 | 7 | 9 | 6 | 6 | 6 | 4 | 2,3333 | 0,6666 | F2–F3 | 2 (lb, ln) | Anterior | Elliptical (worn) | Anastomosed w/protrusions |
| PIMUZ A/III 0823 e | 43 | 37 | 10 | 9 | 8 | 9 | 7 | 4 | 4,1111 | 0,7777 | F2 | 2 (lb, ln) | Premaxilla | Lemon shaped | Reticulate w/ protrusions |
| PIMUZ A/III 0823 f | 42 | 38 | 9 | 8 | 4 | 6 | 7 | 3 | 4,75 | 0,875 | F1 | 0 | – | Elliptical | Smooth |
| PIMUZ A/III 0823 g | 31 | 29 | 7 | 6 | 3 | 6 | 5 | 3 | 4,8333 | 0,8333 | F4 | 3 (ln, m, d) | Maxilla | Lemon shaped (faint) | Smooth (w/ apicobasal striations) |
| PIMUZ A/III 0823 h | 26 | 26 | 5 | 6 | 5 | 4 | 4,5 | 3 | 4,3333 | 0,75 | F1–F2 | 1 (ap) | – | Elliptical | Smooth |
| PIMUZ A/III 0823 i | 21 | 21 | 5 | 5,5 | 3 | 3 | 4 | 2 | 3,8181 | 0,7272 | F2–F3 | 2 (lb, ln) | Anterior dentary | Lemon shaped | Smooth |
| PIMUZ A/III 0823 j | 34 | 34 | 11 | 9 | 8 | 9 | 10 | 4 | 3,7777 | 1,1111 | F4 | 3 (ln, m, d) | Maxilla | Elliptical (worn) | Chevron-like |

the lingual and the apical side of the tooth (F2 abrasion stage). The lingual wear facet has an angle of almost 90 degrees with respect to the labiolingual axis. As no mesial and distal wear facets are present this tooth was probably located anteriorly in the upper jaw (premaxilla). A polished surface is found on the labial side of the crown. Either damage or wear is present on the mesial and distal edges on the carinae, exposing the dentine. The enamel wrinkling pattern is more pronounced on the labial side than on the lingual side of the tooth (see Figs. 4A–4B). On the labial as well as the lingual side, the pattern is more pronounced in the middle of the tooth, and fades out slightly toward the apex and the base. The labial enamel wrinkling pattern consists of frequently anastomosing, sinuous grooves and crests of varying width with a general apicobasal orientation. Grooves and crests are discontinuous; crests are often interrupted by pits and islets. The crests are rounded to triangular in shape. The distribution of crests and grooves is roughly equal. Compared to the other teeth, excepting BSPG 1993 IX 331B, the crests protrude sharply, and the grooves are relatively deep. On the lingual side, more pits are present, and the grooves and crests appear slightly less rounded in shape, but retain their apicobasal orientation. The grooves appear more shallow on the lingual side.

*BSPG 1993 IX 331B (Fig. 2B).* This unworn tooth has the same colour and enamel texture as BSPG 1993 IX 331A. It curves towards the lingual side, with the labial side slightly more convex on the upper half, resulting in a labiolingually tapering apex. The tooth crown is convex toward the distal side, tapering to a mesiodistally narrow apex. The tooth is generally distally inclined, however the apex curves slightly towards the mesial side. It has an oval cross section at the base, which becomes ''lemon-like'' (sensu *Díez Díaz, Tortosa & Loeuff, 2013*) at the apex due to the presence of pronounced carinae on the mesial and distal edges. Its SI is 2.61 and the CI 0.77, see Table 1. The carina on the distal side is slightly more pronounced and continues further basally than the one on the mesial side. The apex contains a polished surface on the mesial side.

The wrinkling is similar to BSPG 1993 IX 331A, however, due to the unworn state of BSPG 1993 IX 331B, it is more pronounced. The enamel wrinkling pattern consists of sharply protruding, angular, narrow and discontinuous grooves (see Figs. 4C–4D). The wrinkling on the lingual side is more pronounced than the labial side, and also appears slightly more rounded. Although it is fully developed, the crown lacks wear facets, and the root does not seem to have any resorption, as in the F1 abrasion stage proposed by *Wiersma & Sander (2016)*. This, together with the coarse enamel pattern of the crown (Figs. 4C–4D), with no signs of abrasion by occlusion, indicates that this was probably a recently erupted (unused) tooth. Due to this, we cannot hypothesize the placement of this tooth in the jaw. This specimen differs from the other teeth in this study by the strong curvature of the crown, and its low SI ratio. This can be attributed to a distal (posterior) placement of the tooth in the jaw, as for example is seen in the nearly complete tooth row of *Giraffatitan* (MB.R.2181.21), *Sarmientosaurus* (*Martínez et al., 2016*), and *Abydosaurus* (*Chure et al., 2010*). In addition, and because of its size, it probably belonged to a juvenile individual.

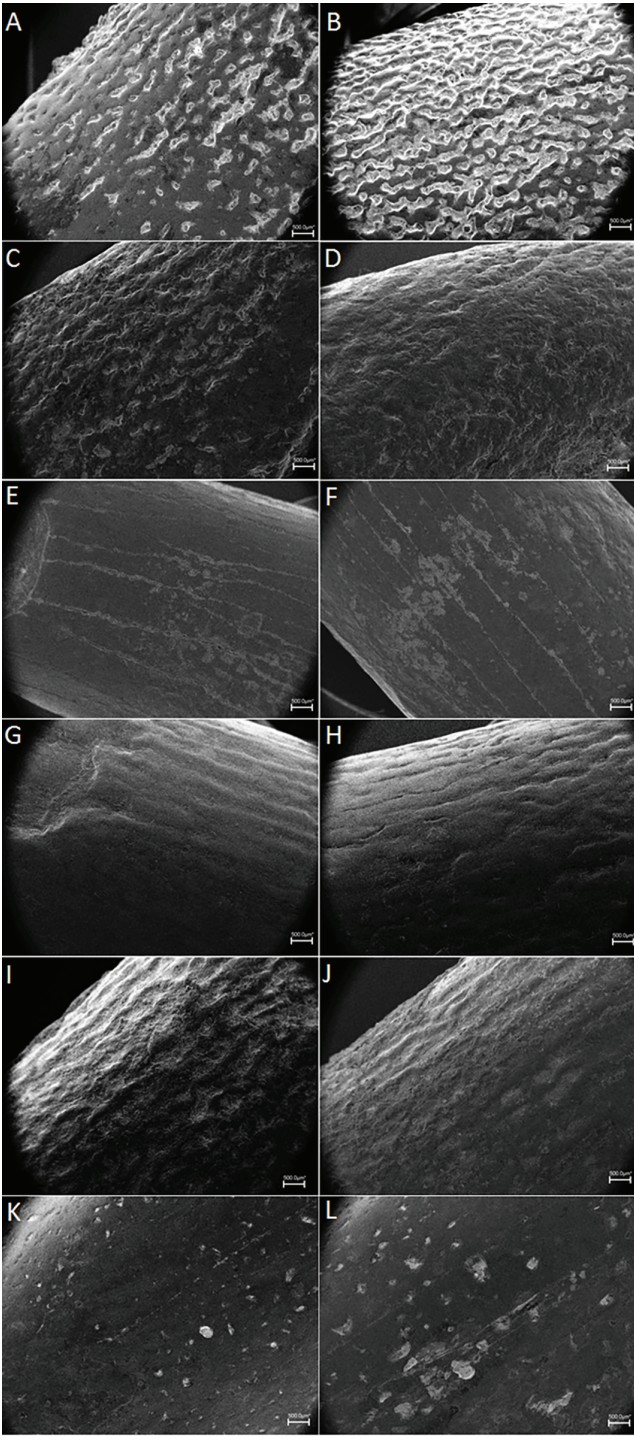

**Figure 4  SEM pictures of enamel wrinkling.** BSPG 1993 IX 331A in labial (A) and lingual (B) view, BSPG 1993 IX 331B in labial (C) and lingual (D) view, BSPG 1993 IX 313A in labial (E) and lingual (F) view, BSPG 1993 IX 2A in labial (G) and lingual (H) view, BSPG 1993 IX 2B in labial (I) and lingual (J) view, and BSPG 1993 IX 2D in labial (K) and lingual (L) view. The scale bar equals 500 μm. Images taken by FH and RM.

*BSPG 1993 IX 331C (Fig. 2C).* This chocolate to reddish-brown coloured tooth differs from the first two Kem Kem teeth, in that it is rather straight both labially and lingually, shows little tapering and has a distinctive enamel wrinkling pattern. The apex shows tapering due to labial and lingual wear facets, and only a slight apical mesiodistal tapering is present. A slight curvature towards the lingual side is present. The base and middle of the tooth is oval in cross section; the apex shows a slightly more elliptical shape. It has a SI of 4.11 and a CI of 0.83, see Table 1.

Four wear facets are present on the tooth, one on each of the labial, lingual, mesial, and distal surfaces, respectively. The lingual wear facet is angled at around 60 degrees with respect to the labiolingual axis. The labial wear facet is angled at almost 90 degrees with respect to the labiolingual axis. Both the mesial and distal wear facets are more pronounced on the lingual side, and are positioned basal to the lingual wear facet. They appear almost parallel to the tooth's main axis. The number and development of the wear facets indicate that this tooth is between the F4 and F5 abrasion stages, with an important tooth-to-tooth contact. The food-to-tooth contact seems to not be as important, as the crown enamel is not as worn as in other specimens of this sample (see e.g., BSPG 1993 IX 313A). Due to the higher development of the lingual wear facet and the placement of the mesial and distal ones, this tooth was probably located in the maxilla.

The enamel of BSPG 1993 IX 331C is ornamented with thick mesiodistally oriented sharply protruding ridges and relatively wide grooves. At about two thirds of the apicobasal height, the grooves slope towards the base from the mesial and distal edges at an angle of about 45 degrees, and meet in the midpoint. Towards the base, the ridges become more horizontally positioned. Some grooves seem to be connected, forming a chevron-like morphology. This wrinkling pattern is also seen in PIMUZ A/III 0823j. Some pits are visible in the apex, probably due to a diet with some grit content.

The four wear facets of BSPG 1993 IX 331C are not seen in similar shapes in any of the other teeth. Moreover, the deep grooves of the enamel wrinkling on the labial and lingual sides of the tooth do not resemble any taphonomic patterns as described by *King, Andrews & Boz (1999)*, but this does not rule out taphonomic processes completely. However, the peculiar pattern is also seen in PIMUZ A/III 0832j, therefore, it is probably natural.

*BSPG 1993 IX 313A (Fig. 2D).* The upper half of this chocolate-brown coloured, worn tooth crown is inclined towards the lingual side, with the labial side curving convexly, and the lingual side curving concavely. The apex tapers mesiodistally, and the mesial side of the tooth inclines distally, creating a convex mesial apical end. The tooth has an oval cross-section at the base, becoming more "lemon-like" (sensu *Díez Díaz, Tortosa & Loeuff, 2013*) towards the apex due to the presence of protruding carinae on the mesial and distal edges. The carina on the distal edge continues further towards the base than the carina on the mesial edge. Its SI and CI are 3.35 and 0.76, respectively, see Table 1.

The apex contains one wear facet on the lingual side, angled at around 75 degrees with respect to the labiolingual axis of the tooth. The presence of only one wear facet, the occlusal one, groups this tooth within the F2 abrasion stage. Due to this, this tooth was probably placed anteriorly in the upper jaw (premaxilla).

The enamel appears smooth, except for thin apicobasally-oriented discontinuous grooves, (see Figs. 4E–4F).

### Moroccan sample: Erfoud

*PIMUZ A/III 0823a (Fig. 3A).* This reddish-brown worn tooth crown is apicobasally straight in lingual and labial view. Only at the apex, in labial view, does the tip taper very slightly to the distal side. In both mesial and distal views, the tooth is seen to curve towards the lingual side, showing a moderate convexity on the labial side, and an equally moderate concavity on the lingual side; also seen in BSPG 1993 IX 331A. The width at the base is only slightly higher than at the middle of the crown, showing a moderate mesiodistal tapering towards the apex. The cross-section of the tooth is oval to elliptical at the base, to oval to possibly lemon-shaped at the apex. The carinae are not pronounced on this tooth, therefore the apical lemon-shaped cross-section is not clear. The carinae on both sides of the tooth appear worn. This tooth has a SI value of 4.92 and a CI value of 0.58, see Table 1.

Labiolingual tapering towards the apex is caused by the presence of a round to oval lingual wear facet (meaning a F2 abrasion stage). This could mean that this tooth could have been a premaxillary one.

The enamel wrinkling pattern is not well-preserved, possibly due to abrasion of the tooth. Only faintly reticulate wrinkling is seen on mostly the labial side of the tooth. This pattern could be a worn enamel wrinkling type similar to that of the more unworn BSPG 1993 IX 331A and BSPG 1993 IX 331B. No protruding ridges or deep prominent grooves are present in the enamel wrinkling pattern of this tooth.

*PIMUZ A/III 0823b (Fig. 3B).* This worn, grey to beige coloured tooth displays a unique morphology amongst the tooth sample, as it is a relatively apicobasally short, mesiodistally wide tooth, with a D-shaped cross-section, representing more a general non-neosauropod eusauropod tooth shape (though this feature also exists in basal titanosaurs; *Barrett & Upchurch, 2005*; *Holwerda, Pol & Rauhut, 2015*). In labial and lingual view, the tooth is apicobasally straight at its lower half, showing a similar mesiodistal width at the base as at the middle. The tooth then tapers mesiodistally at its upper half, and also tapers slightly towards the distal side. Moreover, in lingual view, the upper half of the crown, towards the apex, is slightly mesiodistally constricted, giving the tooth a pear-shaped appearance. In mesial and distal view, the tooth curves towards the lingual side, creating a convex labial and concave lingual side. The cross-section is oval at the base, to D-shaped at the apex. The SI is 2.38 and the CI 0.69, see Table 1.

Both carinae are worn, exposing dentine on both sides, and a prominent oval wear facet is present on the lingual side (F2 abrasion stage). This tooth could have been a premaxillary one. Another worn surface is present on the apex of the labial surface, where the enamel is worn away to expose the dentine.

The enamel wrinkling consists of rugose, protruding, continuous anastomosing ridges, which are flattened by wear and/or abrasion throughout. Between these ridges, discontinuous grooves and pits are visible.

*PIMUZ A/III 0823c (Fig. 3C).* The crown of this well-preserved, unworn tooth is black, whilst the root is cream-coloured to beige. The crown is more or less apicobasally straight in labial and lingual view, however a faint convexity is seen on the medial side, showing a curvature of the entire tooth, from root to apex, towards the distal side. Towards the apex the crown tapers sharply, creating a triangular apical tip. In mesial and distal view, there is only a faint convexity visible on the labial side, whereas the lingual side remains apicobasally straight. The cross-section at the base is oval, and the cross-section at the apex is lemon-shaped (sensu *Díez Díaz, Tortosa & Loeuff, 2013*). It presents a SI of 3.78 and a CI of 0.67, See Table 1.

As no wear facets are present, this tooth has a F1 abrasion stage. No more information about its placement in the jaw can be deduced. Both carinae are well-preserved as sharply protruding ridges, where the distal carina is more pronounced than the mesial, and shows a faint sinusoidal curvature. Furthermore, the distal carina shows ridges inclining from the apicobasally oriented wrinkling towards the carina, creating denticle-like structures, or pseudodenticles (*Bonaparte, González Riga & Apesteguía, 2006*). The enamel wrinkling consists of rugose, discontinuous, sinusoidal, rounded protrusions and islets, which are intersected by shallow but broad anastomosing grooves and pits. The enamel is most pronounced on the lingual side, whilst the labial side shows a polished surface on the upper half of the crown, where the wrinkling is worn away. This tooth was probably used before it was lost, despite the lack of wear facets, because of its labial polished surface (indicating food-to-tooth contact).

*PIMUZ A/III 0823d (Fig. 3D).* This reddish to chocolate-brown, worn and polished tooth crown is apicobasally straight in labial and lingual view. In labial view, the tooth is more rectangular in shape, whilst in lingual view, the mesiodistal width increases from the base towards the middle, after which it tapers towards the apex. The apex of the crown inclines slightly towards the distal side, especially in lingual view. In labial view, two grooves are present, the distal of which is slightly displaced towards the middle of the tooth. In mesial and distal view, the labial side is mildly convex, and the lingual side is more apicobasally straight. The cross-section from base to apex remains similar, being oval to elliptical, without protruding carinae on mesial and distal sides. The SI is 2.33 and the CI 0.67, see Table 1.

Symmetrical, oval wear facets are present on the lingual and labial apices. The presence of both labial and lingual wear facets in the same tooth is an interesting feature that has not been considered in the five abrasion stages (*Wiersma & Sander, 2016*). *García & Cerda (2010)* and *García & Cerda (2010b)* described several apical wear patterns from a sample of titanosaurian teeth from Argentina, and the morphology of the crown and wear facets of PIMUZ A/III 0823d is highly similar to the tooth MPCA-Pv-55 (fig. 5.4). The apex of these teeth (with both labial and lingual apical wear facets) is usually straight and perpendicular to the apicobasal axis of the crown, and the inclination and development of both wear facets is normally different. *García & Cerda (2010)* and *García & Cerda (2010b)* suggested that the presence of both a labial and a lingual wear facet on the same tooth could be related to the position of the teeth on the tooth row, the occlusal pattern, and the replacement of the

opposite teeth (see *García & Cerda, 2010*; *García & Cerda, 2010b*), fig. 7, for a more graphic explanation of this hypothesis). The absence of mesial and distal wear facets suggests that it was probably anteriorly located in the dentary or (pre)maxilla (it is not possible to be more accurate, as both wear facets are equally developed). The mesial and distal sides of the apex do not taper, which further adds to the rectangular appearance of the tooth. The mesial and distal carinae are worn down to smooth, polished ridges, however, the dentine is not exposed.

The enamel wrinkling pattern is mostly polished away by (food-to-tooth or taphonomic) abrasion, and only at the base of the labial and lingual sides a faint wrinkling is present. This consists of rounded, anastomosing protrusions, giving a pebbly textural shape, with wide but shallow grooves and pits in between. At the base of the carinae, the wrinkling becomes slightly more rugose, showing angular protrusions angling at ∼30 degrees with respect to the apicobasal axis, towards the carinae, with deep wide grooves running parallel to these.

*PIMUZ A/III 0823e (Fig. 3E).* The crown of this greyish-brown tooth tapers gently towards the apex in labial and lingual view, and inclines slightly towards the distal side. In labial and lingual view, the tooth crown is apicobasally straight. In mesial and distal view, the tooth is labially strongly convex, but lingually only mildly concave. The apex tapers to a point labiolingually. The cross-section at the base is elliptical to cylindrical, whilst the apex is more D-shaped to elliptical. It has a SI of 4.11 and a CI of 0.78, see Table 1. Both carinae are worn, showing marginal grooves where dentine almost appears through the enamel.

A large oval wear facet is present on the lingual apical side. However, the labial side also shows a less pronounced wear facet, symmetrically placed to that of the lingual side. This tooth was probably placed anteriorly in the upper part of the mouth (possibly premaxilla), as can be deduced from the more developed lingual wear facet and the absence of mesial and distal facets. The poor development of the labial wear facet indicates that its lower new opposite tooth probably erupted only shortly prior to its loss. We consider a F2 abrasion stage to be appropriate for this tooth, due to the poor development of the labial wear facet, and its placement in the premaxilla (because of the absence of mesial and distal wear facets).

The enamel wrinkling pattern is finely reticulate, with a symmetrical distribution of small rounded protrusions surrounded by thin grooves. However, at the base of the crown, the enamel shows deep continuous grooves interspersed with pits, whilst the protrusions are worn away. Part of the enamel at the root is damaged or gone.

*PIMUZ A/III 0823f (Fig. 3F).* This black, smooth, tooth crown shows traces of red sediment. In labial and lingual views, the lower half of the tooth crown is straight, after which it shows a strong inclination towards the distal side, giving the mesial apical half a convex, and the distal apical half a straight to concave, shape. In mesial and distal views, the tooth is rather apicobasally straight, only a faint labial convexity and lingual concavity is seen. The cross-section at the base is round to elliptical, and more cylindrical to a very faint lemon-shape (due to the carinae) at the apex. Apically, the tooth tapers to a sharp tip

which is inclined distally. The SI is 4.75 and the CI is 0.88. Both mesial and distal carinae show a prominent but smooth ridge protruding towards mesial and distal sides on the apical half of the crown, the lower half of the crown does not show carinae.

As no apical wear facets are present we consider this tooth to show a F1 abrasion stage. Due to this, the placement of this tooth in the snout cannot be hypothesized. The enamel is completely smooth, suggesting, together with the lack of clear wear facets, that this tooth was relatively unused and unworn.

*PIMUZ A/III 0823g (Fig. 3G).* This grey to reddish brown tooth crown follows the same general morphology as PIMUZ A/III 0823f, in that the lower half of the crown is apicobasally straight in both labial and lingual views, as well as in mesial and distal views. The upper half of the crown bends towards the distal side, providing the mesial side with a convex curvature. The apex tapers to a sharp tip in labial and lingual view, as well as in mesial and distal view. Both mesial and distal carinae are present, which are prominently but smoothly protruding at the apical half of the crown. The cross-section is elliptical at the base to strongly lemon-shaped at the apex. This tooth has a SI value of 4.83 and a CI value of 0.83, see Table 1.

An apicobasally-elongated, mesiolabially-oriented wear facet, as well as an equally elongated, distallingual wear facet is present, giving the apical half of the crown a more constricted shape, and creating a faint buttress at the base of the mesial and distal constriction. Finally, a faint, small, round apical wear facet is present. The presence of these three wear facets place this tooth in a F4 abrasion stage, as the apex is already gently rounded. This tooth probably was placed in the maxilla. The slight displacement of both mesial and distal wear facets indicate that this tooth was rotated. The enamel is completely smooth, however, apicobasal striations in the enamel are present on the labial apical side of the tooth.

*PIMUZ A/III 0823 h (Fig. 3H).* The smallest tooth of the sample, this reddish-brown crown shows a similar morphology to PIMUZ A/III 0823f and PIMUZ A/III 0823g. The crown is straight in labiolingual and mesiodistal views, with only the mesial apical end inclining towards the distal side in labiolingual view. The apex tapers to a sharp tip. The cross-section is elliptical to cylindrical at the base and at the apex. The SI is 4.33 and the CI is 0.75, see Table 1.

The tooth is completely smooth and nearly unworn; only a small, circular, apical wear facet is present (F1–F2 abrasion stage). This information is not sufficient to propose its likely placement in the snout. Due to its small size and unworn state, this tooth might originate from a juvenile.

*PIMUZ A/III 0823i (Fig. 3I).* This rectangular, red tooth is the second smallest of the Erfoud sample. It has a rectangular shape, showing almost no tapering in labial and lingual view, and only a slight apical tapering in mesial and distal view. In mesial and distal view, the labial side is mildly convex and the lingual side is mildly concave. The cross-section is oval at the base to oval to lemon-shaped at the apex. It has a SI of 3.82 and a CI of 0.73, see Table 1.

The labial apical side shows a high-angled, oval wear facet, whilst a low-angled elongated wear facet is present on the lingual side, which stretches to halfway down the apicobasal length of the crown. This tooth presents the same condition as PIMUZ A/III 0823d, with both labial and lingual apical wear facets. As the apical labial wear facet is more developed, this tooth may have been placed in the lower jaw. The absence of mesial and distal wear facets suggests that it was probably anteriorly located in the dentary. The enamel of the tooth is completely smooth.

*PIMUZ A/III 0823j (Fig. 3J).* This dark brown to black tooth tapers gently towards the apex from the base, whilst being slightly mesiodistally constricted at around the middle. The labial side is convex, whilst the lingual side remains apicobasally straight. Both mesial and distal carinae are present as smooth ridges which do not protrude. The cross-section is elliptical, both at the base as well as at the apex. It presents a SI value of 3.78 and a CI value of 1.11, see Table 1.

The upper half of the tooth crown is polished. However, only on the lingual side, a low angled wear facet is present. Both at the mesial and distal apical sides, a low-angled, oval, polished surface is present as well, indicating tooth overlapping (F4 abrasion stage, as the crown apex is rounded). The location and development of these three wear facets indicate that this tooth was probably located in the maxilla.

The enamel wrinkling pattern is only visible at the base of the tooth, as, similar to BSPG 1993 IX 331C, there is a mesiodistally-positioned, chevron-like pattern of deep wide grooves and ridges, which disperse towards the carinae.

### Algerian sample

*BSPG 1993 IX 2A (Fig. 2E).* The tooth is covered in black enamel, and has a reddish-brown base. The crown is apicobasally straight. The mesial and distal sides taper strongly towards the apex, whereas the labial and lingual sides taper slightly. The mesial side, however, shows a slight convexity, and curves towards the distal side, which is straight apically. The tooth is mesiodistally widest at the middle, after which it tapers towards the apex. Carinae are present on the mesial and distal edges of the upper third of the crown, as smooth protruding ridges. The carina on the distal side shows a slightly sinusoidal curvature at about halfway of the apicobasal length of the tooth, which is also seen in PIMUZ A/III 0823c. The distal carina reaches slightly further basally. The cross-section at the base is oval, becoming lemon-like at the apex due to the carinae (sensu *Díez Díaz, Tortosa & Loeuff, 2013*). The SI is 4.35 and the CI 0.78, (see Table 1).

A wear facet is present on the lingual side, angled at around 50 degrees with respect to the labiolingual axis. The presence of only one wear facet, the occlusal one, groups this tooth within the F2 abrasion stage. As this facet is located lingually, and there are no mesial and distal wear facets, this tooth was probably placed in the maxilla.

The apical part of the labial side also shows a polished surface, as in PIMUZ A/III 0823c. The enamel wrinkling pattern on both the labial and lingual sides consists of apicobasally-oriented grooves and ridges, which are less broad and more sinuous on the labial side than on the lingual side (see Figs. 4G–4H). While unworn, was probably

similar to the crown enamel of BSPG 1993 IX 2B (Figs. 4I–4J), but it has been worn by tooth-to-food contact (like BSPG 1993 IX 313A, Figs. 4E–4F). The enamel on the labial side is slightly thicker than on the lingual side.

*BSPG 1993 IX 2B (see Fig. 2F).* This worn tooth is grey to beige in colour. The tooth is curved lingually, with the labial side convex, and the lingual side concave. This curvature increases towards the apex. A very slight mesiodistal tapering can be seen at the apical part of the crown, however the tooth appears rectangular when observed from the labial and lingual side. Labiolingual tapering appears due to the presence of a labial and lingual wear facet intersecting at the apex. The lingual side of the tooth is relatively flat in the mesiodistal direction when compared to the labial side. This gives the tooth an oval cross section at the base, becoming D-shaped toward the centre, and more cylindrical to lemon-shaped apically due to the presence of carinae. Carinae are present on the mesial and distal edges, although the distal carina is more pronounced and continues further basally. It has a SI of 3.65 and a CI of 0.71, (see Table 1).

Two distinct wear facets are present on BSPG 1993 IX 2B, one on the labial and the other on the lingual side. The labial wear facet is larger (almost 15 mm in apicobasal length), and angled toward the mesial side. The labial wear facet is angled at around 72 degrees with respect to the labiolingual axis of the crown. The lingual wear facet is smaller. It cuts the labiolingual axis at almost 90 degrees. This tooth presents the same condition as PIMUZ A/III 0823d and PIMUZ A/III 0823i, with both labial and lingual apical wear facets. As the apical labial wear facet is more developed in BSPG 1993 IX 2B, this tooth may have been placed in the lower jaw. The absence of mesial and distal wear facets suggests that it was probably anteriorly located in the dentary.

The enamel wrinkling pattern consists mainly of apicobasally-oriented, sinuous grooves and ridges. The ridges are sharply protruding and are triangular in shape (see Figs. 4I–4J). The enamel wrinkling pattern of BSPG 1993 IX 2B appears quite similar to BSPG 1993 IX 331A, BSPG 1993 IX 331B, and the labial side of BSPG 1993 IX 2A. The enamel wrinkling of BSPG 1993 IX 331A and of BSPG 1993 IX 331B is slightly more pronounced than the enamel wrinkling of BSPG 1993 IX 2B, a difference perhaps caused by greater wear on the latter. A noteworthy difference between the enamel wrinkling patterns of this tooth and all other teeth from this sample is that the enamel wrinkling is more pronounced on what seems to be the lingual side, instead of the labial side.

*BSPG 1993 IX 2C (Fig. 2G).* This grey to beige coloured tooth is badly preserved and largely covered with reddish sediment. The crown is fairly straight apicobasally, and tapers apically (mesiodistally as well as labiolingually). The mesial apical side shows a curvature towards the distal side, as seen in BSPG 1993 IX 2A. From the base to the middle, the width of the crown expands slightly labiolingually, after which it tapers to the apex. This expansion mainly seems to occur on the convex (probably labial) side, and is also seen in BSPG 1993 IX 2A. The distal edge shows faint traces of a carina but that cannot currently be accurately determined. The cross-section at the base is slightly oval, becoming almost

circular at the middle of the crown, and then becoming more lemon-like apically due to the presence of carinae. The SI is 4.09 and the CI is 0.87, (see Table 1).

A possible wear facet can be seen on the labial side, cutting the labiolingual axis of the crown at a low angle (35 degrees), so it was probably at a F2 abrasion stage. The absence of mesial and distal wear facets, and the labial position of the one present, indicate that this tooth probably belonged to the anterior part of the dentary. Because of the damaged state of this tooth, no SEM pictures were taken.

*BSPG 1993 IX 2D (Fig. 2H).* This reddish-brown coloured crown tapers mesiodistally towards the apex, as well as labiolingually, resulting in a sharp apical tip. The labial side is convex, whilst the lingual side is more concave. The upper third of the crown shows stronger mesiodistal tapering on the mesial side. Distinctly protruding carinae are present on the mesial and distal edges of the crown. The mesial carina is slightly more distinct due to the curvature of the mesial apical part, however, it only runs halfway along the tooth towards the base, whereas the distal carina continues further basally, to the upper 3/4th of the tooth. The tooth has an oval cross section at the base, becoming strongly lemon-shaped apically due to the distinct carinae. It presents a SI of 3.62 and a CI of 0.71, (see Table 1).

Wear facets are not present (F1 abrasion stage), so its position on the snout cannot be deduced. The enamel is smooth except for some pits (see Figs. 4K–4L). This feature is interesting, not only because this tooth is the one that has the most worn crown enamel, but also for the presence of pits on it. The individual probably fed on low vegetation, with a significant quantity of grit. This tooth had clearly been used for feeding, but occlusion features are not present, so it was probably located anteriorly in the snout, with its opposite tooth unerupted throughout its functional life.

## DISCUSSION

### Systematic discussion and comparisons

The tooth sample from northwestern Africa shares the presence of mesial and distal margins extending parallel to each other along almost the entire length of the crown with neosauropod (diplodocoid and titanosauriform) sauropods, together with the absence of a mesiodistal expansion at the base of the crown (*Calvo, 1994*; *Upchurch, 1995*; *Upchurch, 1998*; *Salgado & Calvo, 1997*; *Wilson & Sereno, 1998*; *Upchurch & Barrett, 2000*; *Barrett et al., 2002*; *Wilson, 2002*; *Upchurch, Barrett & Dodson, 2004*). These teeth also share with diplodocoids and titanosaurs the loss of some plesiomorphic features of Sauropoda, which are retained in some, but not all, basal titanosauriforms (e.g., *Giraffatitan*, *Euhelopus*), such as the presence of a lingual concavity with a median ridge, and labial grooves (*Upchurch, 1998*; *Barrett et al., 2002*).

The general crown outline is similar in all the teeth of the sample: parallel-sided crowns showing slight labiolingual compression with mesial and distal carinae, which express a higher amount of protrusion in BSPG 1993 IX 331B, BSPG 1993 IX 2A, BSPG 1993 IX 2D, PIMUZ 0823c and PIMUZ A/III 0823 h. The labiolingual compression (seen in apical view) is more conspicuous in the Algerian teeth, though PIMUZ A/III 0823a from Erfoud, Morocco, also shows this amount of compression. BSPG 1993 IX 331B and BSPG 1993

IX 2D, PIMUZ A/III 0823d, PIMUZ A/III 0823 h, PIMUZ A/III 0823i present a different crown morphology from the rest of the sample (see below). However, this difference in the crown morphology between these teeth and the rest of the sample could be due to different positions in the tooth row, as occurs in most eusauropods and other sauropodomorphs (e.g., *Carballido & Pol, 2010*; *Chure et al., 2010*; *Holwerda, Pol & Rauhut, 2015*; *Martínez et al., 2016*; *Mocho et al., 2016*; *Carballido et al., 2017*; *Wiersma & Sander, 2016*). Lingual buttresses seen in titanosauriforms from the Lower Cretaceous of Japan (*Barrett et al., 2002*), South Korea (*Lim, Martin & Baek, 2001*), in *Giraffatitan Janensch, 1935*, and in *Astrodon* (*Carpenter & Tidwell, 2005*), or circular bosses as seen in Euhelopus (*Barrett & Wang, 2007*; *Poropat & Kear, 2013*) a possible macronarian from the Upper Jurassic of Portugal (*Mocho et al., 2016*), and an unknown (possible euhelopodid) titanosauriform from the Barremian of Spain (*Canudo et al., 2009*) are not seen in any of our Moroccan or Algerian tooth samples. Neither do the Algerian and Moroccan tooth samples show mesial and distal buttresses such as in an isolated tooth from the Santonian of Hungary (*Ösi, Csiki-Sava & Prondvai, 2017*). Moreover, diplodocoid pencil-like or needle-like teeth as in *Limaysaurus* (*Calvo & Salgado, 1935*; *Salgado et al., 2004*) are not found in this sample either. In addition, two further enamel types can be found in the sample: rugosely or reticulately wrinkled (BSPG 1993 IX 331A, 331B, 331C, PIMUZ A/III 0823a,b,c,e, BSPG 1993 IX 2B and 2C) or smooth (BSPG 1993 IX 313A, PIMUZ A/III 0823d,f,g,h,i, BSPG 1993 IX 2A, and 2D) enamel. The differences between more rugose and more reticulate wrinkling in the enamel ornamentation could be due to the wear of the tooth and the diet of the individual animal.

Finally, although the CI is similar in both morphotypes, the SI range is wider in the Moroccan Erfoud sample, whilst remaining within a similar range between the Moroccan Taouz and the Algerian sample.

After the morphological descriptions and similarities, two main morphotypes could be distinguished, based on shared features of shape, expression of wear facets, carinae and enamel wrinkling. One morphotype, however, may harbour further submorphotypes, which are also discussed, however, without further sampling this could not be proven valid. It should also be noted that the morphotypes proposed here might not always reflect biological morphotypes. The morphotypes are here discussed and compared to other sauropods, including, but not restricted to, biogeographically similar and contemporaneous sauropods (e.g., of Spain, France, northwest and central Africa).

**Morphotype I: BSPG 1993 IX 331A, BSPG 1993 IX 331B, BSPG 1993 IX 313A, BSPG 1993 IX 2A, BSPG 1993 IX 2C, BSPG 1993 2D; PIMUZ A/III 0823a, PIMUZ A/III 0823b, PIMUZ A/III 0823c, PIMUZ A/III 0823e, PIMUZ A/III 0823f, PIMUZ A/III 0823g, PIMUZ A/III 0823h**

*General*

This is the most abundant morphotype in the sample. It consists of teeth with high SI (2,2-4,9) and CI (0,6-0,9), rugose enamel wrinkling (but see discussion on this), prominent mesial and distal carinae, together with a labial convexity, and a slightly distal inclination of the apex, and a subcylindrical to lemon-shaped cross-section. There are

many intra-group specific morphological differences within this morphotype, however as teeth both differ morphologically within one toothrow, as well as between upper and lower toothrows (*Sereno & Wilson, 2005*; *Wilson, 2005*; *Zaher et al., 2011*; *Holwerda, Pol & Rauhut, 2015*; *Martínez et al., 2016*; *Mocho et al., 2016*; *Wiersma & Sander, 2016*), enamel wrinkling as well as the presence of carinae are taken as the main drivers for comparisons (see *Carballido & Pol, 2010*; *Holwerda, Pol & Rauhut, 2015*). Moreover, enamel wrinkling can be demonstrated to change over ontogeny, with indications that juveniles have smooth, or smoother enamel wrinkling in comparison with adult animals (*Fiorillo, 1991*; *Fiorillo, 1998*; *Díez Díaz et al., 2012*; *Díez Díaz, Pereda Suberbiola & Sanz, 2012*; *Díez Díaz, Ortega & Sanz, 2014*; *Holwerda, Pol & Rauhut, 2015*). As stated above, this abundant Morphotype I arguably still hosts several (sub)morphotypes, which are also discussed below.

### Discussion

BSPG 1993 IX 331A and BSPG 1993 IX 331B (Figs. 2A, 2B) show a different SI, and different CI. However, they share a similar morphology in terms of expression of carinae, labial convexity, and enamel wrinkling pattern, and most likely belong to the same morphotype. Similarly, PIMUZ A/III 0823a shares the same apicobasal elongation of BSPG 1993 IX 331A, as well as a worn-down version of the rugose and highly sinuous enamel wrinkling. This reticulate, worn-down version of the rugose type of enamel wrinkling is further found in BSPG IX 1993 313A, BSPG IX 1993 2B, as well as in PIMUZ A/III 0823b and PIMUZ A/III 0823e. BSPG 1993 331A resembles a large Campanian-Maastrichtian titanosaur tooth from Río Negro, Argentina (*García, 2013*), in having a similar crown-root transition, as well as having high CI and SI ratios (0.73 vs 0.85 and 3.73 vs 4.36 for the tooth described by *García (2013)* and BSPG 1993 IX 331A respectively). However, the enamel wrinkling patterns between these two teeth show great differences; the enamel of BSPG 1993 IX 331A shows highly sinuous patterns not visible on the Argentine tooth. The tooth also resembles the cylindrical morphotype with circular cross-section of the southeastern French Campanian-Maastrichtian Fox-Amphoux-Métisson morphotype (FAM 03.06, 03.11, and 04.17; *Díez Díaz, Pereda Suberbiola & Sanz, 2012*, fig. 9). The SI ratios are high in both the Algerian/Moroccan and the French sample (~4, excepting BSPG 1993 IX 331B). These teeth also display a similar labial convexity, which becomes stronger towards the apex in both the Algerian/Moroccan and the French sample. The enamel wrinkling differs between these two, however, as the Kem Kem teeth show a much more pronounced enamel wrinkling. The apicobasal elongation, slight labial convexity, and presence on most of these teeth of an apically based labial and lingual wear facet, also resembles that of the Late Cretaceous Mongolian *Nemegtosaurus* (*Wilson, 2005*). Finally, the moderate tapering (or lack thereof) resembles that of teeth of the Late Cretaceous *Alamosaurus* from the USA (*Kues, Lehman & Rigby, 1980*), although the enamel wrinkling on *Alamosaurus* is smoother than in the Moroccan/Algerian sample, though apicobasal striations are present on both.

Even though the reticulate enamel wrinkling could be a worn-down version of the rugose wrinkling described for this morphotype, BSPG 1993 IX 313A, PIMUZ A/III 0823b and PIMUZ A/III 0823e (Figs. 2D, 3B, 3E) slightly deviate from the general shape of this morphotype in displaying a mesiodistally constricted apex in labial and

lingual view. A difference in order of tooth row might explain this, as teeth from the mesial side of the toothrow are larger and more robust than teeth from the distal end of the toothrow. This is not seen in titanosaurs (*García & Cerda, 2010*; *García & Cerda, 2010b*), however, it is observed in titanosauriforms (e.g., *Giraffatitan* MB.R.2181.21; *Abydosaurus*, (*Chure et al., 2010*). This shape is further seen in more robust (lower SI, higher mesiodistal and labiolingual width) titanosauriform morphotypes, such as the Cenomanian titanosauriform indet. tooth from France, described by *Vullo, Neraudeau & Lenglet (2007)*, *Astrodon* (*Carpenter & Tidwell, 2005*), the more robust tooth form of the Late Cretaceous Asian *Mongolosaurus* (*Mannion, 2011*), *Europatitan* from the Barremian-Aptian of Spain (*Torcida Fernández-Baldor et al., 2017*), the posterior maxillary teeth of the Cenomanian-Turonian Patagonian *Sarmientasaurus* (*Martínez et al., 2016*), *Ligabuesaurus* from the Aptian-Albian of Neuquen, Argentina (*Bonaparte, González Riga & Apesteguía, 2006*), an unnamed titanosaurian tooth sample from the Upper Cretaceous Bissekty Formation, Uzbekistan (*Averianov & Sues, 2017*), and lastly, the robust types of the Massecaps and *Ampelosaurus* morphotypes from the Campanian-Maastrichtian of southeastern France (*Díez Díaz, Tortosa & Loeuff, 2013*). Finally, BSPG 1993 IX 313A, with its strong apical labial convexity, is morphologically similar to both the small tooth type of the Cenomanian-Campanian Chinese *Huabeisaurus* (*D'Emic et al., 2013*), as well as the small Campanian-Maastrichtian *Atsinganosaurus* tooth type from France by sharing a labiolingual compression (CI: 0.76), a high-angled apical wear facet, and lemon-shaped cross section due to the presence of apical carinae (*Díez Díaz, Tortosa & Loeuff, 2013*, fig. 3, MHN-AIX-PV.1999.22). It also matches *Huabeisaurus* and *Atsinganosaurus* in having smooth enamel. One hypothesis for this smooth enamel could be the tooth-to-food contact, that could have worn a previously more coarse enamel, like the ones of BSPG 1993 IX 331A and BSPG 1993 IX 331B (Figs. 4A–4D). However, the enamel of BSPG 1993 IX 313A does not present the conspicuous pits that appear on the enamel of BSPG 1993 IX 2D (Figs. 4K–4L) from the Algerian sample. This could be due to a diet with a greater quantity of grit in the Algerian individual, as happens in many sauropods, especially the ones that fed in the lower levels of the trees (*Fiorillo, 1998*; *Upchurch & Barrett, 2000*; *García & Cerda, 2010*; *García & Cerda, 2010b*). The individual of BSPG 1993 IX 313A may have fed on soft vegetation, as has been suggested for *Diplodocus* (*Fiorillo, 1998*).

Also included in Morphotype I are BSPG 1993 IX 2A, BSPG 1993 IX 2C, PIMUZ A/III 0823c (Figs. 2E, 2G, 3C). These teeth deviate from the general cylindrical and general apical tapering shape of Morphotype I in that they all display a mesiodistally slender base, after which the width increases towards the middle, and then tapers towards the rounded apical tip. This is also arguably seen in *Rinconsaurus* (*Calvo & González Riga, 2003*), as well as in an unnamed titanosaur tooth sample from the Upper Cretaceous Bissekty Formation, Uzbekistan (*Averianov & Sues, 2017*), although the latter teeth show a lingual median ridge, which the Moroccan and Algerian samples do not. BSPG 1993 IX 2A and BSPG 1993 IX 2C display a similar low angled lingual wear facet (∼50 degrees relative to the labiolingual tooth axis), and both BSPG 1993 IX 2A and PIMUZ A/III 0823c show a similar labial polished surface. BSPG 1993 IX 2C is too worn, however the carinae of the well-preserved BSPG 1993 IX 2A and PIMUZ A/III 0823c are prominently present, as mesially and distally

offset ridges. Moreover, the distal carina is more developed than the mesial one in both, showing a sinusoidal curvature in distal view, as seen to a lesser extent in *Huabeisaurus* (*D'Emic et al., 2013*). Note, however, that the carinae are not as prominent as in *Euhelopus* (*Barrett & Wang, 2007*; *Poropat & Kear, 2013*). Furthermore, the distal carina of PIMUZ A/III 0823c shows pseudodenticles, a feature seen in the Late Jurassic South African *Giraffatitan* (*Janensch, 1935*), the Lower Cretaceous Patagonian *Ligabuesaurus* (*Bonaparte, González Riga & Apesteguía, 2006*), an unnamed titanosauriform from the Early Cretaceous of Denmark (*Bonde, 2012*), the Late Jurassic French *Vouivria* (*Mannion, Allain & Moine, 2017*), and the Early Cretaceous African *Malawisaurus* (*Gomani, 2005*).

Furthermore, BSPG 1993 IX 2A resembles the morphotype B of Lo Hueco, Spain (MCCM-HUE 2687, *Díez Díaz, Ortega & Sanz, 2014*; Fig. 5). Both morphotypes are cylindrical, have a high SI (>4.3), a strong apical distal or medial inclination of the tooth (giving the tooth a far from straight outline) with a high-angled wear facet. Moreover, the enamel wrinkling of BSPG 1993 IX 2A matches that of the Lo Hueco morphotype, in that both morphotypes show coarse, but not rugose, discontinuous wrinkling, with smooth longitudinal ridges, although BSPG 1993 IX 2A shows more pronounced enamel wrinkling than Lo Hueco morphotype B.

PIMUZ A/III 0823c also resembles Lo Hueco morphotype A, in the general morphology of the crown and its length, its labiolingual compression, and its SI and CI values (see e.g., HUE-685, *Díez Díaz, Ortega & Sanz, 2014*, fig. 2).

It could be argued that, due to the differences of BSPG 1993 IX 2A, BSPG 1993 IX 2C, PIMUZ A/III 0823c from the rest of Morphotype I, these teeth could be gathered in a separate morphotype, however, as the sample size is low, with only three teeth, further analysis to help distinguish any (sub)morphotypes is not possible and lies outside of the scope of the current study, until a higher tooth sample can be examined. Therefore, although these teeth are discussed as potentially separate, the most parsimonious conclusion is to leave them in Morphotype I.

Finally, BSPG 1993 2D, PIMUZ A/III 0823f, PIMUZ A/III 0823g, PIMUZ A/III 0823 h are assigned to Morphotype I, although the enamel wrinkling differs from the rugose type (both the worn and unworn expression of it). BSPG 1993 IX 2D, PIMUZ A/III 0823f, PIMUZ A/III 0823g and PIMUZ A/III 0823 h (Figs. 2H, 3F, 3G, 3H) all share a smooth enamel type, a strong distal displacement of the apex in labial and lingual view, an acutely tapering, sharp apical tip, and a distally inclined base of 45 degrees. One tooth (PIMUZ A/III 0823h) is likely to be a juvenile tooth. This is interesting, as thus far teeth of juvenile sauropods show a difference in expression of enamel wrinkling, as found by *Fiorillo (1991)*; *Fiorillo (1998)*, *Díez Díaz et al. (2012a)*, *Díez Díaz, Pereda Suberbiola & Sanz (2012)*, *Díez Díaz, Ortega & Sanz (2014)*, and *Holwerda, Pol & Rauhut (2015)* in several titanosaurian teeth from the Ibero-Armorican Island. The smoothness of enamel wrinkling in juvenile teeth, therefore, could express differently in adults, further supporting the most parsimonious conclusion of including these last teeth in Morphotype I. *Barrett et al. (2016)* described some juvenile (hatchling or even embryonic) sauropod teeth of titanosauriforms/camarasaurids from the Berriasian of southern France, which match the apical morphology of this Moroccan tooth. However, the Moroccan tooth lacks the basal mesial constriction and the lingual

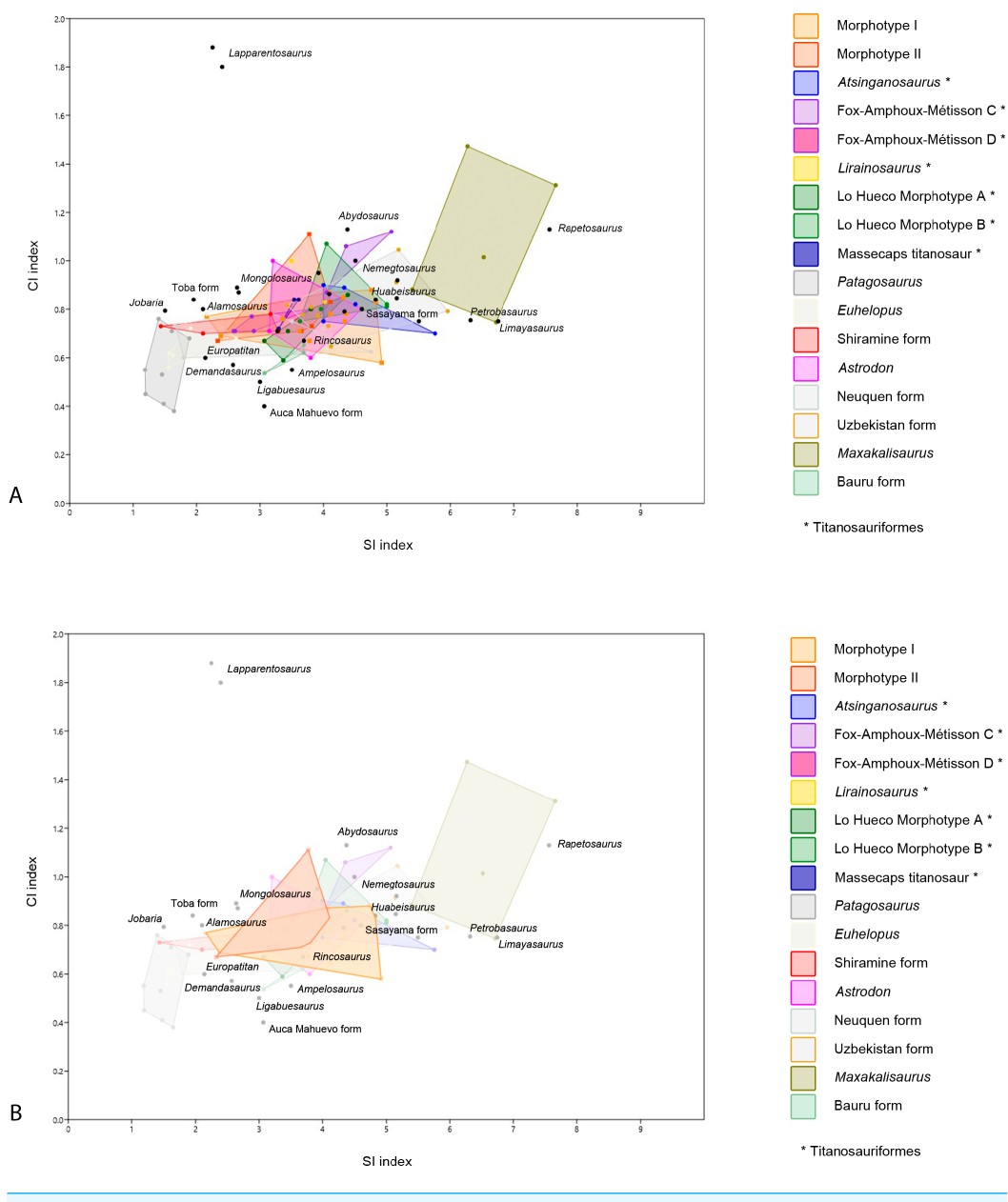

**Figure 5** **Dispersion plot of Cretaceous sauropod tooth morphotypes, with Jurassic outgroups.** See coloured boxes in legend for sauropod groups. (A) All groups highlighted. (B) Morphotype I and II highlighted.

buttress of juvenile teeth described by *Barrett et al. (2016)*. Juvenile teeth are also described by *García & Cerda (2010)* and *García & Cerda (2010b)* for sauropod embryos from Auca Mahuevo, however these small teeth show a more acutely tapering apical tip than the Moroccan sample.

PIMUZ A/III 0823f and PIMUZ A/III 0823 h resemble the Spanish Lo Hueco morphotype A in the general morphology of their crowns and their slight apical asymmetry (which

could be due to wear). PIMUZ A/III 0823f also reflects the Lo Hueco morphotype A in that it has a similar length, labiolingual compression, and SI and CI values (*Díez Díaz, Ortega & Sanz, 2014*). As PIMUZ A/III 0823 h probably belonged to a juvenile specimen, the size of this tooth is smaller when compared to the other ones, and this changes the SI and CI values. Juvenile teeth need to be used with caution when included in comparative studies, especially when using quantitative variables such as the SI and CI (see statistical analysis).

PIMUZ A/III 0823g is the only tooth with smooth enamel that shows striations on the surface, providing the cross-section with a slightly rhomboid shape, however the strongly protruding carinae also still give a lemon-shaped cross section. Striations are seen on teeth of *Huabeisaurus* (*D'Emic et al., 2013*), brachiosaurid teeth from South Korea (*Lim, Martin & Baek, 2001*), smooth enamel type titanosaur teeth from the Bissekty Formation, Uzbekistan (*Averianov & Sues, 2017*), and in combination with smooth enamel, in *Atsinganosaurus* (though some teeth do display gentle wrinkling on the middle, see *Díez Díaz, Tortosa & Loeuff (2013)*, as well as in *Demandasaurus* (*Torcida Fernández-Baldor et al., 2011*), although the carinae are less conspicuous on the latter taxon. Smooth enamel, in combination with a sharp apex, is further seen in the Albian *Karongasaurus* from Malawi (*Gomani, 2005*). Smooth enamel exists on *Limaysaurus* (*Salgado et al., 2004*), however this tooth is more needle-shaped than the teeth of the Moroccan/Algerian sample. As these teeth are not worn, the apical morphology differs from the other teeth of Morphotype I, which are all mostly worn apically. However, as worn versus unworn teeth cannot be properly morphologically distinguished with isolated teeth, these are still assigned to Morphotype I.

Given the resemblance of Morphotype I to predominantly Cretaceous sauropods, and moreover, given the size and shape range, this morphotype is difficult to assign to any particular group. The size and shape range suggests a titanosauriform origin, as derived titanosaurians usually show a more conservative morphological range (*García & Cerda, 2010*; *García & Cerda, 2010b*). However, given the similarity between some teeth of Morphotype I with those of titanosaurians, it is more likely that both non-titanosaurian titanosauriforms as well as titanosaurians are present in the tooth sample of Morphotype I, which reflects sauropod postcranial diversity of the Cretaceous of northwest Africa recorded by previous studies, as both titanosauriforms (*Mannion & Barrett, 2013*; *Fanti, Cau & Hassine, 2014*; *Lamanna & Hasegawa, 2014*) as well as derived titanosaurians (*Ibrahim et al., 2016*) have been reported to be present. Even though some of the teeth resemble more specific groups (e.g., *Euhelopus, Huabeisaurus*) the general morphotype cannot be assigned to any more specific group. No affinity with other groups (e.g., diplodocids, rebbachisaurids) is found for Morphotype I either.

## Morphotype II: BSPG 1993 IX 331C, BSPG 1993 IX 2B, PIMUZ A/III 0823d, PIMUZ A/III 0823i, PIMUZ A/III 0823j

### General

A smaller tooth sample from both the Moroccan and Algerian sample deviates morphologically from Morphotype I. These teeth show a more rectangular morphology, a lack of apical tapering, and distinct labial and lingual wear facets.

## Discussion

BSPG 1993 IX 331C and PIMUZ A/III 0823j (Figs. 2C, 3J) both show a distinctive enamel wrinkling, not seen in the other teeth of the sample. It displays a 'chevron' pattern on the labial and lingual surfaces, with rugose striations anastomosing towards the carinae. This pattern is arguably present on the labial side of the enamel of another Moroccan Kem Kem sauropod tooth, tentatively (but not conclusively) assigned to rebbachisauridae by *Kellner & Mader (1997)*. This latter tooth, however, also shows striations (*Kellner & Mader, 1997*), which are not seen on BSPG 1993 IX 331C, and PIMUZ A/III 0823j. As *Carballido & Pol (2010)* and *Holwerda, Pol & Rauhut (2015)* showed, enamel wrinkling on sauropod teeth can be an autapomorphic diagnostic feature. As both teeth are worn, however, the other morphological features besides enamel wrinkling must be taken into account. The teeth do share morphological similarities in crown shape with the tooth described by *Kellner & Mader (1997)*, which is apicobasally straight. The rectangular morphology of these teeth, caused by a lack of mesiodistal tapering towards the apex, together with a slightly higher CI due to higher labiolingual compression than the other teeth from the Moroccan/Algerian sample, showing an oval to cylindrical cross-section, is shared with BSPG 1993 IX 2B, as well as PIMUZ A/III 0823d and PIMUZ A/III 0823i (Figs. 2F, 3D, 3I). These five teeth further share labial and lingual wear facets, which are symmetrically placed, though in PIMUZ A/III 0823i, as well as in BSPG 1993 IX 2B, one high-angled and low-angled wear facet is seen, which is characteristic of the Aptian-Albian *Nigersaurus* (*Sereno & Wilson, 2005*; p.166, fig. 5.7; *Sereno et al., 2007*, Fig. 2). The size of PIMUZ A/III 0823i matches that of *Nigersaurus*, however, the other teeth are much larger than the isolated *Nigersaurus* tooth described by *Sereno & Wilson (2005)*. The enamel asymmetry, a diagnostic feature for *Nigersaurus*, could not be accurately measured on the Moroccan and Algerian tooth sample. To a lesser degree, the labial and lingual wear facet symmetry is seen in both the Brazilian Aptian *Tapuisaurus* (*Zaher et al., 2011*), as well as the Late Cretaceous *Maxakalisaurus* (*Kellner et al., 2006*; *França et al., 2016*). However, these teeth show a higher SI than the Moroccan/Algerian sample, as well as stronger labial convexity. The morphology of Morphotype II also does not closely match the conical, peg-shaped teeth of *Demandasaurus* (*Torcida Fernández-Baldor et al., 2011*), a rebbachisaurid from the Early Cretaceous of Spain. However, the enamel is relatively smooth on both Morphotype II and *Demandasaurus*, and moreover, the teeth of Morphotype II are unworn, whereas the *Demandasaurus* teeth are relatively unworn.

The morphological features on BSPG 1993 IX 2B further show a set of derived titanosaurian characteristics, with a D-shaped cross-section in the middle of the tooth, as seen in *Mongolosaurus* (*Mannion, 2011*), but also seen in a titanosauriform tooth from Argentina (*García & Cerda, 2010*; *García & Cerda, 2010b*, MPCA-Pv-55, fig. 5). As previously stated, differences in tooth shape can be explained by their relative placement in the toothrow, as seen in *Mongolosaurus* (having both a cylindrical as well as a D-shaped tooth; (*Barrett et al., 2002*; *Mannion, 2011* and *Nemegtosaurus Wilson, 2005*). Both *Mongolosaurus* as well as *Nemegtosaurus*, however, show a distinct tapering of the apex, which BSPG 1993 IX 2B does not show. Furthermore, differences in shape between upper and lower toothrows is also observed in diplodocoids, such as *Diplodocus* and

*Nigersaurus* (*Sereno & Wilson, 2005*). The labial wear facet in particular is characteristic for diplodocids and dicraeosaurids (*Sereno & Wilson, 2005*), and while diplodocids are not known from the Cretaceous, rebbachisaurids and dicraeosaurids are (e.g., *Sereno & Wilson, 2005*; *Gallina et al., 2014*). As rebbachisaurids such as *Rebbachisaurus garasbae* are found in the Kem Kem beds of Morocco (*Lavocat, 1954*; *Mannion & Barrett, 2013*; *Wilson & Allain, 2015*), the Aptian-Albian of Niger (*Nigersaurus* and rebbachisaurids, (*Sereno et al., 1996*; *Wilson, 2005*; *Sereno et al., 2007*), and also the Aptian-Albian of Tunisia (*Fanti, Cau & Hassine, 2014*; *Fanti et al., 2015*), and tentatively in the late Barremian-early Aptian of Spain (Pereda *Suberbiola et al., 2003*; *Torcida Fernández-Baldor et al., 2011*), they are both biogeographically close, as well as originating from beds near-contemporaneous to those of our tooth sample. Morphotype II therefore is tentatively assigned to Rebbachisauridae indet.

### Quantitative analysis

Though quantitative analysis has been applied to theropod teeth in several previous studies (e.g., *Hendrickx & Mateus, 2016*; *Samman et al., 2005*; *Fanti & Therrien, 2007*; *Ösi, Apesteguía & Kowalewski, 2010*; *Hendrickx, Mateus & Araújo, 2015*), a statistical approach to sauropod tooth diversity has thus far only been applied through using one variable—the SI - by *Chure et al. (2010)* and the number of wear facets on the apex (*Averianov & Sues, 2017*). Quantitative analyses on sauropod teeth with two or more variables is therefore an area of study that has not received much attention. Here, we conduct an analysis of sauropod teeth based on multivariate statistical tests. The one-way PERMANOVA revealed significant differences between some of the tested groups ($F = 14.46$, $p = 0.0001$). Specific comparisons for each pair of groups are listed in Table 2. Morphotype I shows significant differences in SI and CI ratios with the D-shaped Morphotype from Fox-Amphoux, *Patagosaurus, Euhelopus,* the Shiramine morphotype, and *Maxakalisaurus* (Table 2). Differences with some of these groups are congruent with the qualitative approach described above. On the other hand, the SI and CI values of Morphotype I did not significantly differ from those of *Atsinganosaurus, Lirainosaurus, Astrodon*, the cylindrical Morphotype from Fox-Amphoux, both tooth morphotypes from Lo Hueco, and the titanosaurian teeth from Massecaps, Neuquén, Uzbekistan, and the Bauru Formation, Brazil (see Table 2). This is in agreement with the qualitative discussion of Morphotype I, where the robust tooth type showed morphological similarities with *Astrodon*, and the elongated cylindrical type was shown to be morphologically comparable to the Neuquén and Uzbekistan tooth types. It also reinforces the morphological similarities found between Morphotype I with Ibero-Armorican titanosaurs.

Concerning Morphotype II, the SI and CI ratios differ significantly from *Atsinganosaurus, Patagosaurus, Euhelopus*, and *Maxakalisaurus* (see Table 2) confirming that even though the wear facets of Morphotype II match those of *Maxakalisaurus*, there is no further morphological overlap. Unfortunately, the available sample size of *Nigersaurus* and other rebbachisaurids was not large enough to statistically test their similarities with confidence (see Table S1).

**Table 2  One-way PERMANOVA comparisons for each pair of sauropod groups.**

| | Morph_I | Morph_II | Atsinga-nosaurus | Fox-Amphoux C-Morph | Fox Amphoux D-Morph | Liraino-saurus | LH Morph A | LH Morph B | Massecaps titanosaur | Patago-saurus | Euhelopus | Shiramine form | Astrodon | Neuquen titano indet | Uzbekistan Neosauropod | Maxakali-saurus | Bauru titanosaur |
|---|---|---|---|---|---|---|---|---|---|---|---|---|---|---|---|---|---|
| *Morph_I*[d] | | 0,3975 | 0,1917 | 0,2584 | **0,0383** | 0,751 | 0,4729 | 0,3556 | 0,4127 | **0,0001** | **0,0002** | **0,0096** | 0,6408 | 0,0774 | 0,2623 | **0,0006** | 0,4152 |
| *Morph_II*[d] | | | **0,0228** | 0,0752 | 0,1809 | 0,8623 | 0,8758 | 0,1156 | 1 | **0,0008** | **0,0038** | 0,1131 | 0,8568 | 0,3702 | 0,0909 | **0,0087** | 0,763 |
| *Atsinganosaurus*[c] | | | | 0,8383 | **0,0083** | 0,3888 | **0,0427** | 0,6971 | **0,0347** | **0,0006** | **0,0022** | **0,0093** | 0,1399 | **0,0495** | 0,6864 | **0,0165** | **0,0357** |
| *FoxAmphoux C-Morph*[c] | | | | | **0,0172** | 0,1892 | 0,0532 | 0,6458 | 0,0954 | **0,0054** | **0,0122** | **0,0268** | 0,1423 | 0,106 | 0,6005 | **0,0176** | 0,1028 |
| *FoxAmphoux D-Morph*[c] | | | | | | 0,1474 | 0,0792 | **0,0143** | 0,144 | **0,0009** | **0,0074** | 0,3581 | 0,1112 | 0,9855 | **0,0057** | **0,0082** | 0,1545 |
| *Lirainosaurus*[c] | | | | | | | 0,6662 | 0,2362 | 0,5032 | **0,0224** | **0,0374** | 0,1364 | 0,9338 | 0,4363 | 0,374 | **0,05** | 0,5989 |
| *LoHueco Morph A*[c] | | | | | | | | 0,0932 | 0,7119 | **0,0015** | **0,0019** | **0,0366** | 0,9081 | 0,321 | 0,0913 | **0,0079** | 0,6607 |
| *LoHueco Morph B*[c] | | | | | | | | | 0,0707 | **0,001** | **0,0024** | **0,009** | 0,189 | 0,0765 | 0,9515 | **0,009** | 0,1212 |
| *Massecaps titanosaur*[c] | | | | | | | | | | **0,0059** | **0,0117** | 0,1099 | 0,7491 | 0,4618 | 0,0936 | **0,0198** | 0,5032 |
| *Patagosaurus*[a] | | | | | | | | | | | 0,0233 | **0,0105** | **0,0021** | **0,0047** | **0,0002** | **0,001** | **0,0061** |
| *Euhelopus*[b] | | | | | | | | | | | | 0,1902 | **0,0051** | 0,0818 | **0,0002** | **0,002** | **0,0128** |
| *Shiramine form*[b] | | | | | | | | | | | | | 0,1433 | 0,5639 | **0,0014** | **0,0086** | 0,1707 |
| *Astrodon*[b] | | | | | | | | | | | | | | 0,3297 | 0,1885 | **0,0077** | 0,6527 |
| *Neuquen titanosaur*[c] | | | | | | | | | | | | | | | 0,0219 | **0,0072** | 0,47 |
| *Uzbekistan Neosauropod*[d] | | | | | | | | | | | | | | | | **0,0008** | 0,0968 |
| *Maxakalisaurus*[c] | | | | | | | | | | | | | | | | | 0,0174 |
| *Bauru titanosaur*[c] | | | | | | | | | | | | | | | | | |

**Notes.**

[a] non-neosauropod eusauropod
[b] non-titanosaurian titanosaurifom
[c] titanosaur
[d] unknown or resolved in this study

However, the very small sample size of this study is not well-suited to this type of approach, therefore the results of this analysis must be taken with caution, and any relationships found between sauropod taxa of our sample must be regarded as tentative. The sauropod relationships suggested by the qualitative and quantitative analysis might not be supported if the sample size is increased. For example, most of the tooth groups of the studied sample were not distinguished after the Bonferroni correction on the post-hoc tests. This might mean that titanosaurian teeth do not possess any clear diagnostic features, or, more likely, that only two variables—the SI and the CI—are not enough to distinguish between titanosaurs, as most taxa and morphotypes strongly group together (see Fig. 5). In Fig. 5, most teeth from the Moroccan/Algerian samples cluster together, and there is also overlap with other sauropod taxa and/or tooth morphotypes. The teeth with high SI and high CI are *Maxakalisaurus, Rapetosaurus, Petrobasaurus* and *Limaysaurus*, whilst teeth with low SI and low CI are *Patagosaurus* and *Jobaria* (see Fig. 5). The taxon with low SI but high CI is *Lapparentosaurus* (Fig. 5). From the central cluster, Morphotype I and II overlap mostly with the Shiramine teeth, the Uzbekistan teeth, *Astrodon,* and the Franco-Iberian teeth (i.e., the Lo Hueco, Massecaps, Fox-Amphoux-Métissons and *Atsinganosaurus* types; Fig. 5). Titanosauriform teeth are known to show a large range in morphology, and, moreover, they show some convergence in morphology with more basal sauropod teeth (*García & Cerda, 2010*; *García & Cerda, 2010b*). However, it is possible, after the qualitative and quantitative analyses here, that titanosaurian teeth show a similar convergence. Finally, as mentioned previously, the large range of size and shape within one toothrow could also cause taxa to accrete in Fig. 5, though postcranial morphology would cause more morphological scatter. To resolve these issues, a larger sample size, not only of the case study at hand, but also of other (Cretaceous) sauropod teeth, could possibly aid in obtaining a better resolution of future quantitative analyses, however this is beyond the scope of the present study.

### Euro-Gondwanan Cretaceous sauropod diversity and palaeobiogeographical implications

To summarize, the described tooth sample from the Albian–Cenomanian Kem Kem beds of Morocco and the Continental Intercalaire of Algeria shows a predominantly non-titanosaurian titanosauriform/titanosaurian assemblage, with only one morphotype (II, with four teeth) showing tentative rebbachisaurid affinities. When compared with the biogeographically nearest tooth assemblages, namely titanosaurian teeth found in the Ibero-Armorican Island, as well as northwest and central Africa, several probable palaeobiogeographical patterns can be assessed.

Morphotype I shows morphological affinities with titanosauriforms. Titanosaurian sauropods in the Campanian Kem Kem beds of Morocco have been reported previously (*Ibrahim et al., 2016*). This titanosaurian predominance is also seen in the later stages of the Cretaceous of Egypt (*Lamanna et al., 2017*; *Sallam et al., 2018*), as well as Spain and France (see e.g., *Le Loeuff, 1995*; *Le Loeuff, 2005*; *Sanz et al., 1999*; *Garcia et al., 2010*; *Díez Díaz et al., 2016*; *Vila, Sellés & Brusatte, 2016*). As fourteen teeth from our sample share several similarities with the cylindrical morphotype from Fox-Amphoux-Métisson,

with the morphotype A from Lo Hueco and with the D-shaped morphotype from Fox-Amphoux-Métisson, as well as with *Atsinganosaurus* and the morphotype B from Lo Hueco (*Díez Díaz et al., 2012*; *Díez Díaz, Pereda Suberbiola & Sanz, 2012*; *Díez Díaz, Tortosa & Loeuff, 2013*; *Díez Díaz, Ortega & Sanz, 2014*), this might suggest a possible close affinity between the Cenomanian North African titanosaurian faunas and those from the Campanian-Maastrichtian of southern Europe.

Morphotype II shows affinities with rebacchisaurids. Rebbachisaurid presence has already been noted from the Early Cretaceous of Niger (*Sereno et al., 1999*; *Sereno & Wilson, 2005*), Tunisia (*Fanti et al., 2013*; *Fanti et al., 2015*; *Fanti, Cau & Hassine, 2014*), and Morocco (*Lavocat, 1954*; *Kellner & Mader, 1997*; *Mannion & Barrett, 2013*; *Wilson & Allain, 2015*), as well as the United Kingdom (*Mannion, Upchurch & Hutt, 2011*) and Spain (Pereda *Suberbiola et al., 2003*; *Torcida Fernández-Baldor et al., 2011*). Again, the North African—southern European connection is a tentative explanation for the dispersion of rebbachisaurids between Gondwana and Europe in the Cretaceous (see Fig. 6); in addition, rebbachisaurids also seem to be a relatively diverse clade within North and Central Africa during the end of the Aptian-Cenomanian (e.g., *Mannion & Barrett, 2013*; *Fanti, Cau & Hassine, 2014*; *Wilson & Allain, 2015*).

In previous studies, faunal connections have been demonstrated to exist between North Africa and Italy, such as temporary continental connections during the Barremian (125 Ma, the so-called Apulian route), or more permanent connections caused by carbonate platforms in the peri-Adriatic (*Gheerbrant & Rage, 2006*; *Canudo et al., 2009*; *Zarcone et al., 2010*; *Torcida Fernández-Baldor et al., 2011*). These last authors suggested that this route could have allowed for the divergence of the rebbachisaurids *Demandasaurus* and *Nigersaurus*. Moreover, *Fanti et al. (2016)* point to a rebbachisaurid dispersal event leading from Gondwana to a European lineage in the Early Cretaceous. Perhaps the hypothesized continental connection also allowed titanosaurs to migrate between Laurasia and Gondwana. Furthermore, next to European sauropods, also theropods, crocodyliforms, amphibians, and snakes and even batoids from southern Europe (France, Spain, Italy, Croatia) have been found to show Gondwanan affinities (*Soler-Gijón & López-Martínez, 1998*; *Gardner, Evans & Sigogneau-Russell, 2003*; *Pereda-Suberbiola, 2009*; *Pereda-Suberbiola et al., 2015*; *Sweetman & Gardner, 2013*; *Csiki-Sava et al., 2015*; *Blanco et al., 2016*; *Blanco et al., 2017*; *Dal Sasso et al., 2016*; *Blanco, In Press*).

The discovery of titanosaurian teeth with similar morphologies from the Cenomanian of Algeria and Morocco, and from the late Campanian-early Maastrichtian of the Ibero-Armorican Island, could indicate that some of these European titanosaurian faunas had a Gondwanan origin. However, this hypothesis needs to be taken with some caution until more postcranial remains (with associated cranial specimens) are found and described from the Early-Late Cretaceous of North Africa and southwestern Europe, as our statistical analysis does not show a high support of any definite grouping between the North African and southern European tooth morphotypes. *Csiki-Sava et al. (2015)* suggested that European titanosaurs do not seem to have a southern influence, and palaeobiogeographical analyses, as well as this study, show both a Gondwanan (South

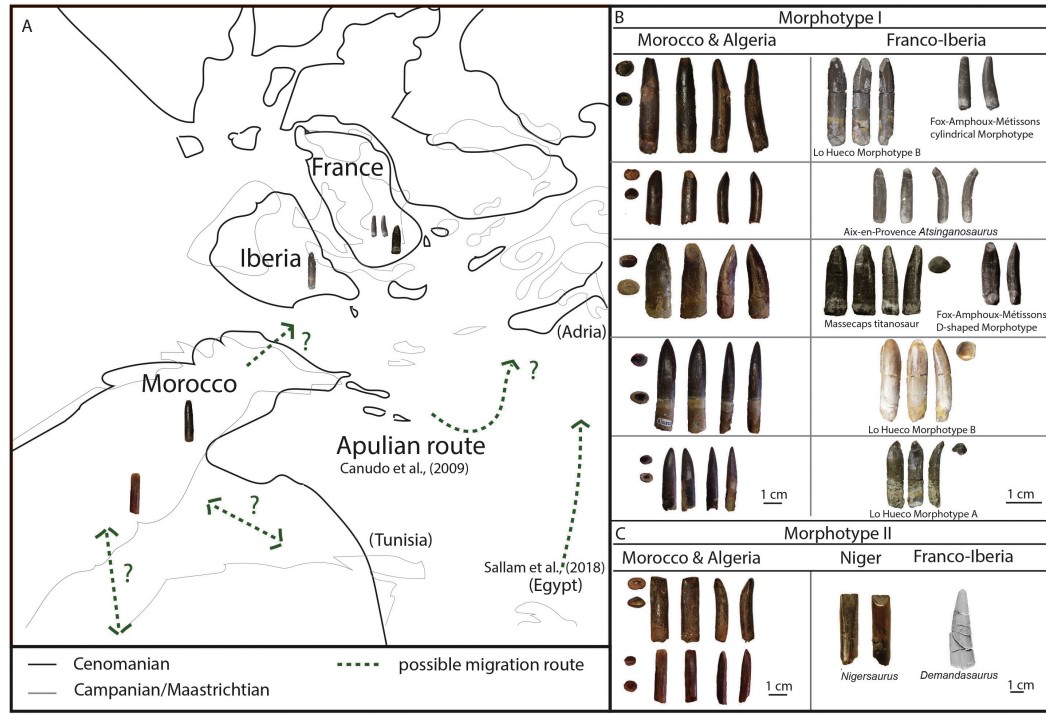

**Figure 6** **Palaeobiogeographical reconstruction of northwest Africa and southern Europe using sauropod tooth morphotypes.** (A) northwest Africa and southern Europe during the Cenomanian (black line) and Campanian-Maastrichtian (grey line), with possible migration routes (dotted green line), adapted from *Csiki-Sava et al., 2015*. (B) Tooth morphotype I next to tooth morphotypes A and B from Lo Hueco (from *Díaz, Ortega & Sanz, 2014*) Fox-Amphoux-Métissons cylindrical type and D-shaped morphotype (from *Díaz, Suberbiola & Sanz, 2012b*), the Massecaps titanosaur and *Atsinganosaurus* (from *Díaz, Tortosa & Loeuff, 2013*). (C) Tooth morphotype II next to *Nigersaurus* from *Sereno et al., 2007*, and *Demandasaurus* (From *Torcida Fernández-Baldor et al., 2011*).

American) and North American input in European Cretaceous fauna (e.g., *Upchurch, 2008*; *Ezcurra & Agnolín, 2011*), which might cautiously be supported by our qualitative and quantitative analyses as well. However, more recent studies have added more information about the North African-southern European connection between these sauropod faunas in the Late Cretaceous (see e.g., *Sallam et al., 2018*; *Díez Díaz et al., 2018*). In Fig. 6, a tentative palaeobiogeographical construction is shown using the Algerian and Moroccan morphotypes and their morphological counterparts from northwest and central Africa, as well as southern Europe, and possible migratory routes based on previous hypotheses.

## CONCLUSIONS

A sample of eighteen teeth from the Cretaceous Kem Kem beds from Erfoud and Taouz, Morocco, and from the Continental Intercalaire, Algeria, has been studied. The overwhelming majority of this sample shows titanosauriform/titanosaur affinities, with a smaller group showing rebacchisaurid affinities. This is congruent with the results of the statistical analyses. However, the small size of the comparative samples and the relative

scarcity of this type of analysis in studies thus far, begs for caution in interpreting results until further study is possible.

On the other hand, similarities between tooth samples from northwestern and central Africa, and southwestern Europe, do strongly hint at a possible sauropod faunal exchange through continental connections in the early Late Cretaceous between North and Central Africa, and between North Africa and southwestern Europe. These results support previous hypotheses from earlier studies on faunal exchange and continental connections.

### Institutional abbreviations

| | |
|---|---|
| **BSPG** | Bayerische Staatssammlung für Paläontologie und Geologie, Munich, Germany |
| **FAM** | Fox-Amphoux-Métisson, France |
| **MBR** | Museum für Naturkunde, Berlin, Germany |
| **MCCM-HUE** | Museo de las Ciencias de Castilla-la Mancha, Spain |
| **MHN-AIX-PV** | Natural History Museum Aix-en-Provence, France |
| **MPCA-Pv** | Museo Provincial "Carlos Ameghino", colección de paleovertebrados, Río Negro, Argentina |
| **PIMUZ** | Palaeontological Institute and Museum, University of Zürich, Switzerland |

## ACKNOWLEDGEMENTS

Oliver Rauhut (BSPG Munich, Germany), Christian Klug and Torsten Scheyer (PIMUZ Zürich, Switzerland) are profoundly thanked for kindly allowing us to use material from their collections. We are indebted to Enrico Schwabe (ZSM Munich, Germany) for his kind assistance in SEM imaging. Many thanks also to Daniela Schwarz and Jens Koch (Museum für Naturkunde Berlin, Germany) for allowing us to use their *Giraffatitan* specimen. We are grateful to Matt Dale and Charlie Underwood for their aid in Kem Kem palaeoecology. Jeff Liston kindly checked the English. The comments and recommendations of editor Kenneth de Baets, and reviewers Paul Barrett, Attila Ösi, and one anonymous reviewer, greatly improved this manuscript. This work was part of the BSc thesis of RM at Utrecht University (the Netherlands).

### Funding

Alejandro Blanco is supported by the program Axudas postdoutorais da Xunta de Galicia 2017—Modalidade A. Additional funding for Alejandro Blanco came from the Synthesys Project DE-TAF-7025 and from the program Axudas á investigación da UDC 2017, 2018. The other authors received no funding for this work. The funders had no role in study design, data collection and analysis, decision to publish, or preparation of the manuscript.

## Grant Disclosures

The following grant information was disclosed by the authors:
Axudas postdoutorais da Xunta de Galicia 2017—Modalidade A.
Synthesys Project DE-TAF-7025.
Axudas á investigación da UDC 2017, 2018.

## Competing Interests

The authors declare there are no competing interests.

## Author Contributions

- Femke M. Holwerda conceived and designed the experiments, performed the experiments, analyzed the data, prepared figures and/or tables, authored or reviewed drafts of the paper, approved the final draft.
- Verónica Díez Díaz performed the experiments, analyzed the data, prepared figures and/or tables, authored or reviewed drafts of the paper, approved the final draft.
- Alejandro Blanco and Roel Montie performed the experiments, analyzed the data, prepared figures and/or tables, approved the final draft.
- Jelle W.F. Reumer conceived and designed the experiments, approved the final draft.

## Data Availability

Raw data consists of Slenderness Index (SI) and Compression Index (CI) of sauropod teeth, mainly from the Cretaceous, with some outgroups from the Jurassic. The dataset is compiled using all literature and personal observations that, to our knowledge, exist of Cretaceous sauropod teeth. This dataset was used for the quantitative analysis to compare Morphotype I and Morphotype II of the sauropod tooth sample from Morocco and Algeria to other sauropods. The Taouz, Morocco, and Algerian tooth sample of eight teeth is stored in the collections of the Palaeontological Museum Munich (BSPG), the Erfoud, Morocco tooth sample of 10 teeth is stored in the collections of the Palaeontological Museum of the University of Zurich (PIMUZ), and both are free and openly accessible upon request to the curators in Munich and Zurich.

## Supplemental Information

Supplemental information for this article can be found online at http://dx.doi.org/10.7717/peerj.5925#supplemental-information.

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
