# Peer review of "Late Cretaceous sauropod tooth morphotypes may provide supporting evidence for faunal connections between North Africa and Southern Europe"

_PeerJ, doi:10.7717/peerj.5925_

## Round 0.1 · original submission · Major Revisions

You thoroughly describe sauropod teeth from North Africa which is of key importance. I would like to see your paper published, but there are some crucial points I would like you to resolve before publication.

Title: Although I feel it is interesting to discuss/speculate about the possible implications for biogeography, I think you cannot be that firm with just 8 teeth to bear on this question (see also comment by reviewer 3). I think describing these teeth is interesting on its own and sufficient to merit publication in peerJ, so I would rather change the title to Sauropod teeth (morphotypes) from North-Africa or something along these lines. I would keep the potential implications for biogeography for the discussion.

Localities/Age assignments: you mention your samples are from Taouz in Algeria – Are you sure about this? There is also a Taouz in Morocco near the Algerian border. Furthermore, the age range for the Kem-Kem beds is quite larger and even larger for the Algerian intercalaire – how can you be sure they were deposited at the some time and contain teeth of the same taxa. It would be appropriate to discuss the age constraints in greater detail (see comment by reviewer 2)

Use of teeth: I am not an expert on sauropod teeth, but I wonder to what degree teeth can be diagnostic for entire taxa. As they are also used for feeding, could some characters not just reflect similarities or differences in feeding. Furthermore, there seems to be some evidence for large intraspecific differences depending on position in tooth row and through ontogeny on the one hand and small interspecific differences on the other hand (see reviewer 1) – in this case, it would be appropriate to at least discuss the range of variation seen in particular species and genera which are better studied.

Quantitative analysis: the paper is currently quite qualitative which is not necessarily an issue. However, I feel the paper would benefit from a more quantitative comparison of these North African teeth with teeth from Europe and other regions as well as those of related groups where teeth could be assigned to particular taxa (see also Reviewer 1) – it would also make it more easy to follow for non-experts. There are several things that could be done – a more phylogenetic analysis (e.g., Cau 2017) or just an ordination/cluster analysis of your data to see their distribution in relation with more well-known taxa from Europe or other regions. Furthermore, when involving more data – a discriminant analysis could potentially also be performed to see if teeth attributed to different taxa can actually be separated with these characters. As you present the measurements of several characters in table this could easily done. Furthermore, I think it would also strengthen your interpretation that teeth are more similar to European one than to those of other regions – similar data should be added for teeth from Europe and other regions for this purposes. If I did cluster or ordination analyses myself on your dataset, teeth number eight often plotted rather together with 2, 3 and 4 rather than with the remaining teeth so one can ask the question if it really belong in the Algerian group. There are several methods which could select how many specimens could be assigned to different groups justified based on the available data (k-means clustering). We did something similar for belemnite morphotypes (e.g., Dera et al. 2016, Fig. 11), but I am sure other more appropriate references should be available on vertebrates.

Faunal connections: I think it would be appropriate to discuss evidence for potential faunal interchanges during the interval not just for dinosaurs (see comment by reviewer 2). Furthermore, some authors have also discussed differences between individual paleogeographic models and it would be appropriate to point larger differences between different models (e.g., Upchurch 2008).

Style: There are several issues with formatting – repeated references in text, missing references from bibliography, etc. Please resolve these issues upon resubmission (see particularly comments by reviewers 1, 2 and 3).

Crucial references missing: reviewers have pointed out that several crucial and relevant are missing (see particularly reviewer 1)

In addition to these and other points raised by the reviewers, please also address my comments in the annotated pdf. These are quite some suggestions and potentially new analyses hence my decision for major revisions. However, i do think these changes/analyses can be easily performed. Please do not hesitate to contact me if you would have additional questions. Looking forward to receiving your revised manuscript.


Suggested references:

Cau A. (2017) Specimen-level phylogenetics in paleontology using the Fossilized Birth-Death model with sampled ancestors. PeerJ 5:e3055 https://doi.org/10.7717/peerj.3055

Dera, G., Toumoulin, A., & De Baets, K. (2016). Diversity and morphological evolution of Jurassic belemnites from South Germany. Palaeogeography, Palaeoclimatology, Palaeoecology, 457, 80-97.

Upchurch, P. (2008). Gondwanan break-up: legacies of a lost world?. Trends in ecology & evolution, 23(4), 229-236.

Reviewer 1 ·

Basic reporting

Dear Editor and authors.

It was a pleasure to review the paper entitled “Titanosaurian tooth morphotypes as supporting evidence of Late Cretaceous landbridges between North Africa and Southern Europe”. I am providing in attachment a pdf with the major part of my comments and suggestions. I am not a native English speaker, so I avoid to make comments about this. The English seems correct, nevertheless, I had some minor doubts in some points of the text. A careful review by the authors should be enough to make sure that everything is ok.

So, my opinion. This paper made me think a little about if we are doing well on this. I understand the need to publish (I am also in the same situation), but sometimes, this definitely reduces the impact of our conclusions and our finally work. I don’t have a clear answer to solve this problem, so it is only a personal reflection. I suppose that this paper appears in the possibility to publish for free on Peer J, if submitted in February, and some of the mistakes that I found herein, might be related with the need to achieve the proposed deadline. If I am correct, the editors should have this in account in future “free opportunities”, because I feel this paper is far to be done for submission. However, it is excellent to have a paper talking about Cenomanian sauropods of Africa, a still pretty unknown group of dinosaurs in this territory, and this is for the MAJOR point of this manuscript.

Major problems:
- Along the text, you lack several important references (some of them from the last five years). I will cite some that for me are almost obligatory, in order to discuss your material: Wilson (2005), Wilson and Allain (2015), Wilson et al. (2016), Mocho et al. (2017), Martinez et al. (2016), Wiersma and Sander (2017), Sereno et al. (2007), Nowinski (1971), Seagusa and Tomida (2011), Gallina and Apesteguía (2011), etc…
- In your systematic discussion you should provide a better discussion with other titanosaurs, especially the ones that have several teeth referred, inclusively from the same skull, such as, Nemegtosaurus, Tapuiasaurus, Sarmientasaurus, Lirainosaurus, etc… The establishment of systematic approaches based on titanosaurian teeth is especially problematic, and you should provide good diagnostic features that allow you to separate them in different groups (a table and figures will be highly recommended with several examples. Only after this you are able to achieve a good discussion).
- Throughout the text, you provide very vague comparisons, like “seems similar”. Avoid this. Systematic phylogeny needs shared features (synapomorphies), diagnostic features, and exclusive set of features, in order to support the established relationships between the Cenomanian taxa and European taxa. I am not asking for a cladistics analyses, but I need to have the diagnostic features in order to accept your hypothesis. A lot of the differences pointed by the authors are also justified in the text with different preservation, positions, diets, etc. So, is difficult to me accept your last conclusion. Because some teeth resembled others, you propose a contact between the sauropod fauna of North Africa and Europe around the Cenomanian.
- Other thing is: some sauropod taxa present an important positional variability on the tooth morphology, and these is well-reported in some taxa such as Camarasaurus or Giraffatitan (taxa that were studied by the authors). The morphology of titanosaurs seems much more conservative, so, is much more difficult to establish different morphotypes, especially different morphotypes with taxonomic signal. In the Late Jurassic of Portugal, for example, you have an important morphological variability of titanosauriform teeth (see Mocho et al. 2017), part of this variability is related with the position and possible ontogeny (if not, the diversity of this clade is definitely underestimated, and at least, 3-5 taxa should be present on the Iberian Peninsula during the Late Jurassic). So the differences that you are observing have a taxonomic value? Maybe yes, maybe not, complete sets of teeth should be used to compare, but a lot of titanosaurian teeth are not possible to differentiate up to the moment. This should be one of your major questions on this paper. Why such a low morphological disparity!? We have characters able to be diagnostic?
- Enamel ornamentation. You have some important papers about this question Holwerda et al. (2015), Mocho et al. (2017) and Wiersma and Sander (2017). You seem to doubt herein on the usage of these features for taxonomic assumption, you should provide more clear position along the text. A same morphotype might shows an important morphological variation of the enamel ornamentation along the tooth, along the tooth row and during ontogeny (see Holwerda et al. 2015, and Mocho et al., 2017). Holwerda et al. (2015) for a fossil record from the same period of time suggest that wrinkled pattern have taxonomic value, identified at least three different sauropods. Mocho et al. (2017), also provided several differences between clearly different morphotypes and taxa, but also identified important variability between teeth of the same morphotype and possible same taxon. So, any taxonomic assumption should be taken with caution, especially with teeth separated by millions of years.
- Different rules are applied during for the references in the text. Check this situation
- Finally, I definitely think, that you should focus your paper on the important information that you can provide for the characterization of the pretty unknown Cenomanian sauropod fauna of Morocco and Algeria. In my opinion, the paleobiogeographic interpretation that you provided lacks strong evidence and mainly supported by the already published evidence of faunal contact between Europe and Africa in the Early and Mid-Cretaceous. You should present evidences for the differentiation between teeth from diplodocoids and derived titanosauriforms. And be very clear about why these teeth should be referred to Titanosauria indet. and not Somphospondyli indet.
I will suggest this paper for publication with major revisions. I will be glad to review a second version.

Experimental design

See above

Validity of the findings

See above

Additional comments

See above

·

Basic reporting

This MS describes eight sauropod teeth from the western part of North Africa. Late Cretaceous dinosaur fossils are extremely rare in Africa, so every piece of a new discovery is important.
The MS is generally well organized, but I have some comments and suggestions that should be clarified before the MS will be published.
English is clear. Literature references should be completed (see comments below).
Article structure is ok.

Experimental design

Research question is well defined, relevant. Methods described with sufficient detail.

Validity of the findings

The only problem is the exact location and age of the specimens.

Additional comments

1) If I understand well the exact locality and thus the age of these specimens are practically unknown. Four of the teeth originate from the Kem Kem area, suggesting a Cenomanian age. On the other hand, the four teeth from Algeria are quite uncertain in age:
„The Continental Intercalaire of Algeria is less studied tan the Kem Kem, and the age ranges from Barremian to Aptian-Albian, however, most authors set the age of the beds close to the Moroccan border, where our Algerian specimens allegedly are from (Taouz, Algeria), to ranging from the Cenomanian to Turonian, with the Cenomanian layers being most fossil rich” – this means to me that these Algerian teeth might even not be Late but Early Cretacoeus in age. So, I would strongly suggest to be more careful to outline paleobiogeographical interpretations based, at least, on the latter four teeth.

2) The authors use the word „landbridge” both in the title and in the MS. As I know there is still no clear evidence for a true land connection between any of the South European landmasses (Iberia, Apulia) and Africa during the Late Cretaceous, thus I suggest to use rather „faunal interchange” (see Buffetaut 1989) of faunal connection phrase instead.
Both in the Introduction and the Discussion parts of the MS the authors discuss a Late Cretaceous faunal connection between the European archipelago and Africa, but refer only a few, relatively recent publications. This hypothesis, however, was already well outlined and supported by many other earlier works, dealing not only with sauropods, such as e.g. Buffetaut (1988, 1989), Vullo et al. (2004), Dalla Vecchia 2002, Pereda-Suberbiola (2009), Rabi and Sebők (2015), Csiki et al. (2015). So, I strongly ask the authors not to forget these workers, who did a lot of work already to demonstrate this scenario, in most cases, based on fossils with more exact age and locality data.

3) Description of the teeth are well done; I have only one comment to this. The authors describe macrowear features (i.e. position, orientation of the wear facets) of the teeth but only once add some info on the microwear details (e.g. scratch, pit orientation, length, proportion). At least the presence or absence of scratches and pits and perhaps scratch orientation would be useful information on those specimens where SEM pictures are available.


4) Citation-reference block is very incomplete! At least a dozen of cited Refs are not in the Reference list, indicated in the annotated pdf (see attached).

5) Figures are well done.

6) Some minor comments and typos have been also indicated in the annotated pdf.

·

Basic reporting

In general the authors provide clear descriptions and their arguments are easy to follow. There are numerous minor grammatical and typographic errors, which I have tried to indicate on the annotated PDF version of their MS.

The authors have cited much of the relevant literature with respect to Kem Kem sauropods and to recent work on relevant sauropod tooth morphology. However, there are large number of referencing errors. The following references are cited in the text, but are not listed in the bibliography:

Calvo & Salgado 1995
Cavin & Forey 2004
Chiappe et al. 2001
De Broin et al. 1971
Erickson 1996
Forey and Cavin 2010
Garcia 2013
Garcia & Cerda 2010
Gheerbrant & Rage 2006
Gomani 2005
Kellner 2006
King et al. 1999
de Lapparent and Gorce 1960
Mannion 2009
Mannion 2011
Rodrigues et al. 2011
Rogers & Forster 2004
Sereno & Wilson 2005
Smith and Dodson 2003

Also, it is difficult to distinguish the papers by Diaz et al. for the years 2012 (two papers) and 2013 (three papers) and the authors should use letter suffixes to make it clearer which paper is being cited.

Finally, Upchurch (1995) and Upchurch (1998) are both duplicated in the reference list (as Upchurch 1995a, b and Upchurch 1998a, b). They are cited in the main text as Upchurch (1995) or (1995a) and Upchurch (1998) or (1998a). This needs correction.

Figures are well produced and show the relevant information, but it would also have been useful to provide images of the tooth microwear as this forms part of the discussion and description.

Most of the raw data are provided, with the exception of more detailed descriptions of the tooth microwear.

Experimental design

The research question is explicit and clearly defined. The authors provide novel data that has not been published elsewhere and use it to address species diversity in the sauropod dinosaur assemblage from the early Late Cretaceous of North Africa. This compliments previous work based on postcranial (largely vertebral) material. The methodologies they employ, based largely on detailed descriptive anatomy, are appropriate and carried out competently. All specimens are deposited in an accredited repository and the authors provide sufficient information for their work to be replicated. Given that sauropod teeth are common in these deposits, it's slightly disappointing that the authors were unable to obtain a larger sample size with which to investigate this question. In addition, although numerous comparisons are provided with European titanosaurs, comparisons with other sauropods are less extensive, which might have biased the authors towards accepting the similarities of their samples to the European material (thus suffering from at least a mild form of confirmation bias in the experimental design).

Validity of the findings

The authors replicate the finding that titanosaurians are present in the sauropod fauna of northern Africa, using a different data set. They also test the frequently made assumption that many of these sauropod teeth are referable to Rebbachisauridae and find it to be lacking as they did not identify any detailed similarities to rebbachisaurid teeth in their sample. However, as their sample is numerically limited I would urge caution in interpretation as a broader sample of these teeth might identify further differences. Also, as noted by the authors, some of the differences seen in their sample are likely due to either preservational differences or positional differences within the tooth row, so I would urge a little more caution in identifying three distinct morphotypes (two seems more defensible). More comparisons with titanosaur teeth from other regions would have been useful as this could have allowed the authors to rule out similarities to taxa from other regions, which could have provided more/less support for the European links that they propose in the MS. Also, some additional basic reporting on the tooth microwear data would have been useful (providing measurements like scratch length, width and orientation frequencies). This in turn could have been used to discuss possible palaeoecological differences between the sauropod taxa, which could provide additional support for the presence of more than one taxon. It would also have been useful to note that there aren't many other African titanosaurs for comparison, so ruling relationships in/out with African taxa isn't really possible at the moment due to sampling, so the implied European connection might simply be the best option at the moment by default. Some nuance in the conclusions would seem to be justified given the small sample size and some of the other ambiguities alluded to above.

Additional comments

NA

---

## Round 0.2 · Minor Revisions

I apologize in the delay of making my decision, but i was still waiting for a review from one of the reviewers. Thank you for increasing your sample size, adding a quantitative analysis of the available data and being more clear on what interpretations are well supported and which ones might be more speculative. The manuscript is easier to follow. There are still some minor points i would like you to take care off (see also annotated pdf):

Formatting: The are still some issues with the formatting of some reference in the text (see annotated pdf for some examples). Particularly when you mention the authors concluded this or that - in this case, their name should not be in brackets, but only the year (please make sure references are properly formatted throughout the text).

Methodology: It would be appropriate to state how you dealt with worn teeth (see comments by reviewer).

Inclusion of more taxa: The reviewer suggested it would be appropriate to add more somphospdylans and diplodocoids, although I think the is not necessarily necessary for the scope of the paper.

Missing references: I agrew with reviewer that for the interpretation on the position and tooth-to-food and tooth-to-tooth contact, adding additional references would be appropriate.

Figures: Please homogenize figure 1 including style of the columns as well as the continent borders (see comments by reviewer). Would it be possible to add indications of features like wear facets, carinae and striations you describe in the text. I like the new figure 5 very much, but it might help to highlight in some way which taxa are titanosauriforms/titanosaurs or rebbachisaurids for readers not (so) familiar with their taxonomy .This is probably best done in the legend, by adding asterisks, other symbols, brackets, etc. or in some other way. It might be appropriate to refer to the route proposed by Sallam et al. (2018) in Figure 6.

Table 1: I agree with the reviewer, that I would be appropriate to add a column for remarks where you refer how you measured teeth, it situ or literature. Furthermore, in the latter case, it would be good to refer the source of these measurements (which table, figure or part in the text).

Table 2: this table would be easier to follow if name behind morphotype 1 and 2 would be order alphabetically. Furthermore, it would also benefit by highlighted significant values in bold.

Please address these and additional comments in the annotated pdfs. Looking forward to receiving your revised manuscript.

Reviewer 1 ·

Basic reporting

Experimental design

Validity of the findings

Additional comments

It was a pleasure to review the paper entitled “Could Late Cretaceous sauropod tooth morphotypes provide supporting evidence for faunal connections between North Africa and Southern Europe?”. I am providing in attachment a pdf with the major part of my comments and suggestions. I am not a native English speaker, so I avoid to make many comments about this, however, this time, the English seems a little confusing in some sentences. The editors and authors should read carefully the text, in order to correct some problems.

The review provided by the authors improved significantly the content of this paper, congrats! However I found some little mistakes that should be corrected (see attachment, and I leave some major comments)

Major problems:

In the morphotype I, I miss a sentence saying why these teeth are not Rebbachisauridae (you have a detailed comparison, but not a final conclusion about it, and also why is not a non-titanosaur somphospondylan). What is the exclusive set of characters only present in Titanosauria (or some of the titanosaurian clades you are studying), and not present in derived somphospodylans like Phuwiangosaurus, Huabeisaurus or Mongolosaurus?
In your quantitative analyses, explain exactly what you did with the worn teeth. If they were included, you should do an analyses without them, and see how they affect your result. I also feel you can provide a better description and discussion of your result. The inclusion of more non-titanosauria somphospodylan titanosaurs and diplodocoids would be good to test better your taxonomic approach, but I also feel that this can be out of scope, taking into account the significantly improvements provided by the authors.
For your interpretation on the position and tooth-to-food and tooth-to-tooth contact, I miss some references. In methodology you just can refer which studies you are using to interpret the position of the tooth (and explain a little how works). One more time, be aware of the exceptions like Nemegtosaurus and Tapuiasaurus,
I have some major problems with the figures.

1 - The figure 1, the columns have different styles, are you using the originals? I suggest you to provide a clean version made by you, and similar styles. The borders of the continents are not acceptable.

2 – in the illustrations of the teeth I miss some indications of the features that you are describing in your manuscript, for example the wear facets, carinae and striations.

3 – In the figure of the paleobiogeography, why you are not refering the route proposed by Sallam et al. (2018)?

In the table you provide as a supplementary material with the measurements, can you had a column for comments and refer how you measured the teeth, if in situ or from the literature. In the ones where the measurements are extracted from the literature, refer precisely which table or figure you used to measure.

I will suggest this paper for publication with moderate revisions. I will be glad to review a third version.

Annotated reviews are not available for download in order to protect the identity of reviewers who chose to remain anonymous.

---

## Round 0.3 · Minor Revisions

Thank you for integrating my suggestions including adding the source of your measurements in the supplementary material. Your manuscript is as good as accepted. I just wanted you to address some minor additional formatting issues before publication:

Line 19: check preferred e-mail address: gmail or uni-muenchen (as listed on the first page)
Line 76: i would rather write the lowest level of taxonomy which has been attempted or possible
Line 874: i would write "conduct" as you did the analysis yourselves
Line 971; i would write "our statistical analysis"
Line 973-979: this sentence is very long and the brackets around "which...well") do not make a lot of sense. Please break up this sentence in at least 2 sentences (before "which ..." or after ".... well".
Figure 5: I feel it would benefit the figure if you would reproduce a duplicate were only the dataset of morphotypes I and II are connected/higlighted just below the current figure 5.
Table 1: I feel it would be possible to fit this table on 1 page. Furthermore, please highlight which taxa are are titanosauriforms/titanosaurs versus rebbachisaurids as this would help the reader not familiar with the higher assignments of these taxa - this could be done by added an additional column and/or symbols.

Looking forward to seeing this published.

---

## Round 0.4 · accepted · Accept

Thank you for implementing my final suggestions and the change made to figure 5 which make it even easier to follow for the reader.

#